# MultiOOD: Scaling Out-of-Distribution Detection for Multiple Modalities

**Hao Dong**[1]   **Yue Zhao**[2]   **Eleni Chatzi**[1]   **Olga Fink**[3]

[1]ETH Zürich   [2]University of Southern California   [3]EPFL

{hao.dong, chatzi}@ibk.baug.ethz.ch, yzhao010@usc.edu, olga.fink@epfl.ch

## Abstract

Detecting out-of-distribution (OOD) samples is important for deploying machine learning models in safety-critical applications such as autonomous driving and robot-assisted surgery. Existing research has mainly focused on unimodal scenarios on *image* data. However, real-world applications are inherently multimodal, which makes it essential to leverage information from *multiple modalities* to enhance the efficacy of OOD detection. To establish a foundation for more realistic **Multi**modal **OOD** Detection, we introduce the first-of-its-kind benchmark, MultiOOD, characterized by diverse dataset sizes and varying modality combinations. We first evaluate existing unimodal OOD detection algorithms on MultiOOD, observing that the mere inclusion of additional modalities yields substantial improvements. This underscores the importance of utilizing multiple modalities for OOD detection. Based on the observation of *Modality Prediction Discrepancy* between in-distribution (ID) and OOD data, and its strong correlation with OOD performance, we propose the Agree-to-Disagree (*A2D*) algorithm to encourage such discrepancy during training. Moreover, we introduce a novel outlier synthesis method, *NP-Mix*, which explores broader feature spaces by leveraging the information from nearest neighbor classes and complements *A2D* to strengthen OOD detection performance. Extensive experiments on MultiOOD demonstrate that training with *A2D* and *NP-Mix* improves existing OOD detection algorithms by a large margin. To support accessibility and reproducibility, our source code and MultiOOD benchmark are available at https://github.com/donghao51/MultiOOD.

## 1 Introduction

Most existing machine learning (ML) models are trained under the closed-world assumption, where the test data is assumed to be drawn *i.i.d.* from the same distribution as the training data, referred to as in-distribution (ID). However, in open-world scenarios, test samples can be out-of-distribution (OOD), thus impacting model robustness and safety [67]. OOD detection aims to detect samples with semantic shifts that are undesirable for the model to generalize [67] and is critical for deploying ML models in safety-critical domains such as autonomous driving [21], robotics [18, 4], and diagnostics for critical assets [24]. Numerous OOD detection algorithms have been developed, ranging from classification-based to distance-based methods [66]. Classification-based methods typically derive confidence directly from the classifier, employing post-hoc processing techniques such as Maximum Softmax Probability (MSP) [31] and Energy [43] or training strategies such as logit normalization [65] and outlier synthesis [22]. Distance-based methods typically measure distances in high-dimensional feature spaces to distinguish between ID and OOD [41, 59]. Additionally, other methodologies explore density estimation [1, 50] and reconstruction techniques [70] for OOD detection.

Current research in OOD detection has predominantly focused on *unimodal* settings, often involving images as inputs [66]. While several recent works [47, 63] have investigated vision-language

38th Conference on Neural Information Processing Systems (NeurIPS 2024).

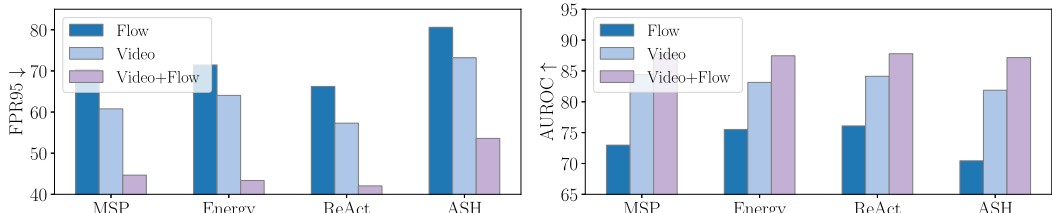

Figure 1: The FPR95 (lower is better) and AUROC (higher is better) on HMDB51 dataset across various modalities. Multimodal OOD substantially improves unimodal OOD w/o bells and whistles.

models [53] to enhance OOD performance, their evaluations are still limited to benchmarks containing *solely images*. Consequently, existing methods fall short in fully leveraging the complementary information from various modalities, such as LiDAR and camera in autonomous driving [21], as well as video, audio, and optical flow in action recognition [56]. To underscore the importance of using multiple modalities in OOD detection, we evaluate representative OOD algorithms across various modalities on the HMDB51 [39] dataset within our MultiOOD benchmark. This is an action recognition task and all models are trained solely using cross-entropy loss between a one-hot target vector and the softmax output. Results in Fig. 1 show that even a simple fusion of video and optical flow modalities can substantially enhance OOD detection performance.

To establish a foundation for more realistic **Multi**modal **OOD** Detection, we introduce a novel OOD benchmark named MultiOOD (Fig. 2), which is the first benchmark for Multimodal OOD Detection and covers diverse dataset sizes and modalities. MultiOOD comprises five video datasets with over $85,000$ video clips in total. The datasets vary in the number of classes, ranging from 7 to 229, and in size, spanning from $3k$ to $57k$. *Video*, *optical flow*, and *audio* are used as different types of modalities. While most existing unimodal OOD algorithms designed for images can be directly applied to Multimodal OOD Detection, such approaches may yield suboptimal results without accounting for the interaction and complementary nature of diverse modalities. As depicted in Fig. 1, the AUROC performance is very close for all unimodal baselines, underscoring the importance of developing OOD detection algorithms tailored to effectively exploit information from multiple modalities.

In this work, we first identify and illustrate the *Modality Prediction Discrepancy* phenomenon on the MultiOOD benchmark, where the discrepancies of softmax predictions across different modalities are shown to be negligible for ID data while significant for OOD data (Fig. 3). We discover a strong correlation between such discrepancies and the OOD detection performance (Fig. 4). Motivated by these observations, we introduce the Agree-to-Disagree (*A2D*) algorithm, which aims to enhance such discrepancies during training. The algorithm is designed so that different modalities should *Agree* on the prediction of the ground-truth class, and *Disagree* on other classes by maximizing the distance between their predictions. Additionally, we propose a novel outlier synthesis algorithm named *NP-Mix*, designed to use the information from nearest neighbor classes to explore broader feature spaces and complement *A2D* to strengthen the OOD detection performance.

Extensive experiments on the MultiOOD benchmark demonstrate the superiority of *A2D* and *NP-Mix*. The integration of *A2D* and *NP-Mix* yields substantial performance enhancements over existing unimodal OOD detection algorithms. For instance, on the UCF101 [57] dataset within MultiOOD, our approach reduces the FPR95 from $32.14\%$ to $10.68\%$ for ASH [16] method, representing a noteworthy absolute $21.46\%$ improvement over the baseline. Our contributions include:

- We highlight the significance of integrating more modalities for OOD detection and introduce MultiOOD, the *first* benchmark for Multimodal OOD Detection encompassing diverse dataset sizes and various combinations of modalities.

- We conduct comprehensive evaluations of existing unimodal OOD algorithms on MultiOOD, revealing their limitations in multimodal scenarios.

- We propose a novel *A2D* training algorithm, inspired by the observation of the *Modality Prediction Discrepancy* phenomenon, alongside a new outlier synthesis algorithm *NP-Mix* that explores broader feature spaces and complements *A2D* to strengthen the OOD detection performance.

- Extensive experiments conducted on MultiOOD underscore the effectiveness of *A2D* and *NP-Mix*. Our source code and MultiOOD benchmark will be made publicly available, facilitating future research endeavors in Multimodal OOD Detection.

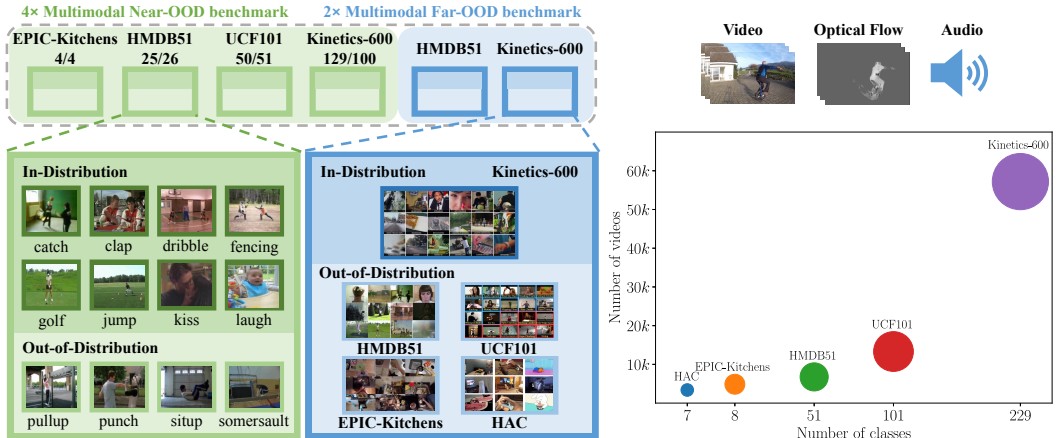

Figure 2: An overview of MultiOOD Benchmark.

## 2 Preliminaries: Multimodal Out-of-distribution Detection

Multimodal OOD Detection aims to detect samples with semantic shifts using ***multiple modalities***. We consider a training set $\mathbb{D}_{in} = \{(\mathbf{x}_i, y_i)\}_{i=1}^n$ drawn *i.i.d.* from the joint data distribution $P_{\mathcal{X}\mathcal{Y}}$, where $\mathcal{X}$ is the input space and $\mathcal{Y} = \{1, 2, ..., C\}$ is the discrete label space. Each training sample $\mathbf{x}_i$ is comprised of $M$ modalities, denoted as $\mathbf{x}_i = \{x_i^k \mid k = 1, \cdots, M\}$. Let $\mathcal{P}_{in}$ denote the marginal distribution on $\mathcal{X}$ and $f : \mathcal{X} \mapsto \mathbb{R}^C$ be a neural network trained on samples in $P_{\mathcal{X}\mathcal{Y}}$ that predicts the label of each input sample. The $f$ in Multimodal OOD Detection comprises of $M$ feature extractors $g_k(\cdot)$ and a classifier $h(\cdot)$. Each feature extractor $g_k(\cdot)$ extracts an embedding $\mathbf{Z}^k$ for its corresponding modality $k$, and the classifier $h(\cdot)$ takes the combined embeddings from all modalities as input and outputs a prediction probability $\hat{p}$:

$$\hat{p} = \delta(f(\mathbf{x})) = \delta(h([g_1(x^1), ..., g_M(x^M)])) = \delta(h([\mathbf{Z}^1, ..., \mathbf{Z}^M])), \tag{1}$$

where $\delta(\cdot)$ is the softmax function. We further include a separate classifier $h_k(\cdot)$ for each modality $k$ to get predictions from each modality separately, with the prediction probability from the $k$-th modality as $\hat{p}^k = \delta(h_k(g_k(x^k)))$.

When deploying $f$ in the real world, it should not only accurately classify known samples as ID, but also identify any "unknown" sample as OOD. A separate score function $S(\mathbf{x})$ is often used to decide whether a sample $\mathbf{x} \in \mathcal{X}$ is from $\mathcal{P}_{in}$ (ID) or not (OOD):

$$G_\eta(x) = \begin{cases} \text{ID} & S(\mathbf{x}) \geq \eta, \\ \text{OOD} & S(\mathbf{x}) < \eta \end{cases},$$

where samples with higher scores $S(\mathbf{x})$ are classified as ID and vice versa, $\eta$ is the threshold. Existing OOD detection studies predominantly focus on unimodal scenarios, with a detailed literature review offered in Appendix A. To establish a foundation for more realistic Multimodal OOD Detection, we introduce a novel MultiOOD benchmark (Sec. 3) and propose an effective multimodal training strategy (Sec. 4) that yields significant enhancements over existing unimodal approaches.

## 3 MultiOOD Benchmark

We create the MultiOOD benchmark to understand the existing gap in Multimodal OOD Detection research. OOD detection primarily focuses on detecting semantic shifts, with two main approaches used for constructing OOD benchmarks. A common method involves considering an entire dataset as in-distribution (ID) and further collects datasets, which comprise similar tasks but are disconnected from any ID categories, as OOD datasets. In this scenario, both semantic and domain shifts are present between the ID and OOD samples. We term this setup as *Far-OOD* in our benchmark. Another approach is to partition the categories of existing datasets into two subsets, referred to as closed (ID) and open set (OOD). Here, both ID and OOD samples originate from the same distribution, with only semantic shifts existing between them. We denote this setup as *Near-OOD* within our benchmark;

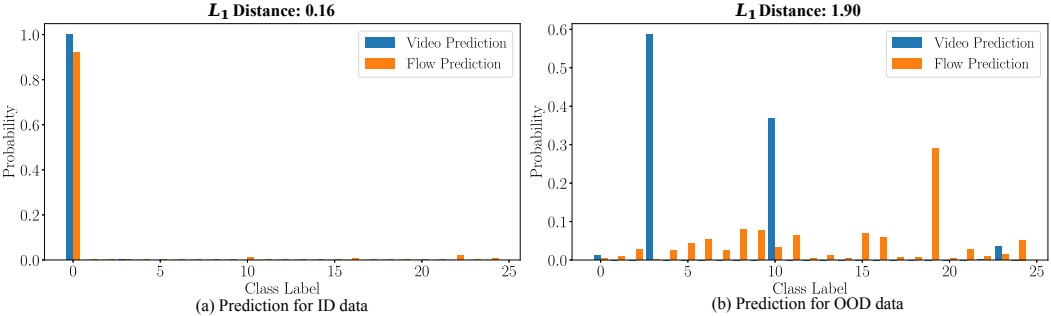

Figure 3: An example of softmax outputs for ID and OOD data. The predictions from video and optical flow demonstrate uniformity across ID data and exhibit variability across OOD data.

a setup that poses greater challenges compared to *Far-OOD*. This setup is also referred to as open set recognition (OSR) in some studies [61, 37, 9]. Notably, OSR and OOD detection both share the same goal of identifying test samples with semantic shifts without compromising the accuracy of ID classification [67]. In our benchmark, we treat OSR and OOD as synonymous concepts and adopt OOD as the general term. MultiOOD comprises five action recognition datasets (EPIC-Kitchens [48], HAC [20], HMDB51 [39], UCF101 [57], and Kinetics-600 [6]) with over $85,000$ video clips in total. The datasets vary in the number of classes, ranging from 7 to 229, and in size, spanning from $3k$ to $57k$. *Video*, *optical flow*, and *audio* are used as different types of modalities. An overview of the MultiOOD benchmark is provided in Fig. 2, with additional details available in Appendix B.

## 3.1 Multimodal Near-OOD Benchmark

In the *Near-OOD* setup, we include four datasets. **EPIC-Kitchens 4/4** is derived from the EPIC-Kitchens Domain Adaptation dataset [48], where the dataset is partitioned into four classes for training as ID and four classes for testing as OOD, with a total of $4,871$ video clips. Similarly, **HMDB51 25/26** and **UCF101 50/51** are constructed based on HMDB51 [39] and UCF101 [57], with a total of $6,766$ and $13,320$ video clips respectively. In the case of **Kinetics-600 129/100**, we select 229 action classes from the Kinetics-600 dataset [6], with each class comprising approximately 250 video clips and a total of $57,205$ video clips. Within this setup, 129 classes are designated for training as ID, while the remaining 100 classes are allocated for testing as OOD.

## 3.2 Multimodal Far-OOD Benchmark

In the *Far-OOD* setup, we include HMDB51 and Kinetics-600 as ID datasets.

**HMDB51 dataset as ID.** For the OOD datasets, we utilize UCF101, EPIC-Kitchens, HAC, and Kinetics-600 datasets. All of these datasets are carefully curated to remove samples that belong to ID classes in HMDB51. Given the relatively small number of classes in the EPIC-Kitchens and HAC datasets, we remove 8 classes in the HMDB51 dataset that overlap with EPIC-Kitchens and HAC, with 43 classes left as ID classes. For UCF101, we remove 31 overlapping classes with HMDB51, resulting in 70 classes designated as OOD classes for evaluation. For other datasets, no class overlap exists and we utilize their original classes as OOD.

**Kinetics-600 dataset as ID.** Similarly, we adopt UCF101, EPIC-Kitchens, HAC, and HMDB51 datasets as OOD datasets, with careful selection undertaken to exclude samples belonging to ID classes in Kinetics-600. We carefully selected a subset of 229 action classes from Kinetics-600 in the *Near-OOD* setup to mitigate the potential overlap with other datasets. For the UCF101 dataset, we remove 11 overlapping classes with Kinetics-600, leaving 90 classes as OOD classes for evaluation. For other datasets, there are no class overlap issues and we use their original classes as OOD.

## 4 Methodology

In this section, we first identify the *Modality Prediction Discrepancy* phenomenon on the MultiOOD benchmark and demonstrate its substantial correlation with the OOD detection performance (Sec. 4.1). Subsequently, we introduce the Agree-to-Disagree (*A2D*) algorithm aimed at enhancing such discrep-

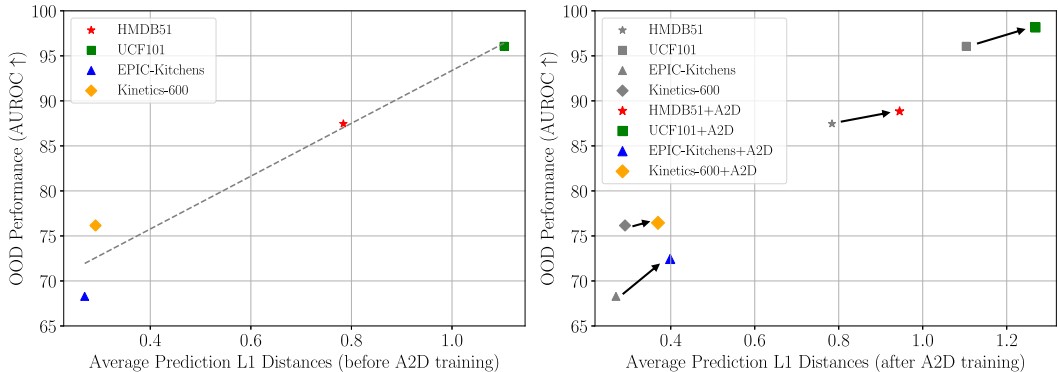

Figure 4: Average prediction $L_1$ distances between ID and OOD data ($l_{OOD} - l_{ID}$) before and after *A2D* training across various datasets within the MultiOOD, where Energy [43] is used as score function. The distances are highly correlated to the ultimate OOD performance. *A2D* training amplifies such distances, consequently enhancing the efficacy of OOD detection.

ancies during training (Sec. 4.2). Finally, we propose the novel outlier synthesis algorithm named *NP-Mix* that complements *A2D* to further strengthen the OOD detection performance (Sec. 4.3).

## 4.1 Modality Prediction Discrepancy

We first explore the predictive behaviors demonstrated by various modalities when using both ID and OOD data as input on MultiOOD. We compute the prediction probabilities employing classifiers for video and optical flow on the same ID and OOD samples and calculate their $L_1$ distances. As depicted in Fig. 3, for ID data, the prediction probabilities of both video $\hat{p}^1$ and optical flow $\hat{p}^2$ are generally exhibited consistent with each other on the ground-truth label, consequently yielding a small $L_1$ distance $\|\hat{p}^1 - \hat{p}^2\|_1$ between them. Conversely, for OOD data, each modality tends to express varying confidence preferences towards distinct classes, resulting in a notable increase in the $L_1$ distance between their predictions. We refer to this phenomenon as *Modality Prediction Discrepancy* between ID and OOD data. This discrepancy can be attributed to the unavailability of semantic information on OOD data during model training, stimulating each modality to generate conjectures based on its unique characteristics upon encountering OOD data during testing.

To verify the universality of this phenomenon, we calculate the average $L_1$ distance between predictions of video $\hat{p}^1$ and optical flow $\hat{p}^2$ on both ID and OOD data within the HMDB51 dataset. The average prediction $L_1$ distance is 0.63 for ID data ($l_{ID}$) and 1.42 for OOD data ($l_{OOD}$), revealing a substantial discrepancy. In addition, we evaluate other datasets in the *Near-OOD* setup within the MultiOOD benchmark and observe similar discrepancies ($l_{OOD} - l_{ID}$). Such discrepancies have a positive correlation with the OOD detection performance, as illustrated in Fig. 4.

## 4.2 Agree-to-Disagree Algorithm

Motivated by the *Modality Prediction Discrepancy* and its strong correlation with Multimodal OOD Detection performance, we introduce the Agree-to-Disagree (*A2D*) algorithm to foster the amplification of such discrepancies during training. The underlying idea is that different modalities should *Agree* on the prediction regarding the ground-truth class, while they should *Disagree* on the remaining classes by maximizing their prediction distance. *A2D* enables the model to diversify predictions across modalities, consequently yielding high prediction discrepancies for OOD data during testing.

Given a training sample **x** with label $c$, we obtain prediction probabilities $\hat{p}$ from the combined embeddings of all modalities, and $\hat{p}^1$, $\hat{p}^2$ from individual modality, all of which are of shape $[1, C]$, where $C$ represents the number of classes. By removing the $c$-th value from $\hat{p}^1$ and $\hat{p}^2$ (different modalities should *Agree* on the prediction regarding the ground-truth class), we derive new prediction probabilities without ground-truth classes, denoted as $\bar{p}^1$ and $\bar{p}^2$ with shapes $[1, C-1]$. Subsequently, we aim to maximize the discrepancy between $\bar{p}^1$ and $\bar{p}^2$, which can be defined as:

$$\mathcal{L}_{Discr} = -Discr(\bar{p}^1, \bar{p}^2),\tag{2}$$

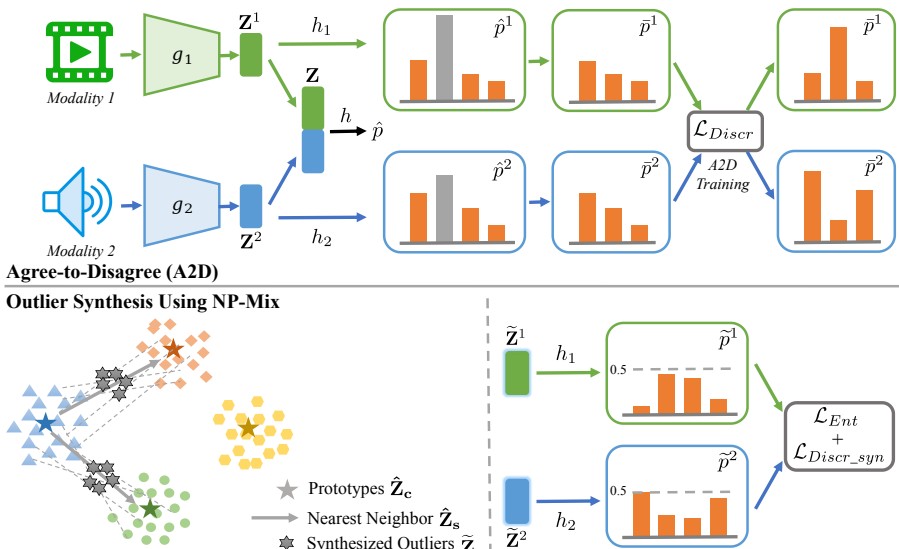

Figure 5: An overview of the proposed framework for Multimodal OOD Detection. We introduce *A2D* algorithm to encourage enlarging the prediction discrepancy across modalities. Additionally, we propose a novel outlier synthesis algorithm, *NP-Mix*, designed to explore broader feature spaces, which complements *A2D* to strengthen the OOD detection performance.

where $Discr(\cdot)$ is a distance metric quantifying the similarity between two probability distributions. We utilize the Hellinger distance [49] and explore the efficacy of alternative distance metrics in our ablation study. The Hellinger distance between two probability distributions is defined as:

$$D(\bar{p}^1, \bar{p}^2) = \sqrt{\frac{1}{2} \sum_{i=1}^{C-1} \left( \sqrt{\bar{p}_i^1} - \sqrt{\bar{p}_i^2} \right)^2}. \tag{3}$$

Furthermore, to ensure that the ground-truth classes possess the highest probabilities, we incorporate the cross-entropy loss $CE(\cdot)$ for each prediction, defined as:

$$\mathcal{L}_{cls} = \frac{1}{3}(CE(\hat{p}, y) + CE(\hat{p}^1, y) + CE(\hat{p}^2, y)). \tag{4}$$

The final loss is obtained as the weighted sum of the previously defined losses:

$$\mathcal{L} = \mathcal{L}_{cls} + \gamma \mathcal{L}_{Discr}, \tag{5}$$

where the hyperparameter $\gamma$ controls the relative importance of the discrepancy term.

### 4.3 Nearest Neighbor Prototype-based Mixup for Outlier Synthesis

Outlier synthesis [22, 60] has demonstrated its efficacy in OOD detection by imposing regularization on the model's decision boundary during training. Introducing the discrepancy loss in *A2D* to the synthesized outlier data for model regularization has the potential to further enhance OOD detection performance. However, existing outlier synthesis methods [22, 60] typically generate outliers near the ID data (Fig. 7), neglecting to explore the broader embedding spaces, thereby potentially leading to suboptimal performance. Inspired by the recent approach introduced in [19], we introduce a novel algorithm termed **N**earest Neighbor **P**rototype-based **Mi**xup (*NP-Mix*), aimed at synthesizing outliers capable of spanning wider embedding spaces by leveraging the information from nearest neighbor classes, as shown in Fig. 5 and Fig. 7. To synthesize outliers, we concatenate the embeddings of all modalities ($\mathbf{Z} = [\mathbf{Z}^1, \mathbf{Z}^2]$) and treat them as a unified entity. Subsequently, we compute a prototype embedding $\hat{\mathbf{Z}}_{\mathbf{c}}$ for each class $c$ by calculating the mean of all embeddings within that class. For each prototype embedding $\hat{\mathbf{Z}}_{\mathbf{c}}$, we identify its top $N$ nearest neighbor prototypes and randomly select one prototype $\hat{\mathbf{Z}}_{\mathbf{s}}$ from them for mixing. Two samples, $\mathbf{Z_1}$ and $\mathbf{Z_2}$, are randomly chosen from class $c$ and class $s$ respectively. The outlier $\widetilde{\mathbf{Z}}$ is generated by their convex combination:

$$\widetilde{\mathbf{Z}} = \lambda \mathbf{Z_1} + (1 - \lambda)\mathbf{Z_2}, \tag{6}$$

Table 1: **Multimodal Near-OOD Detection** using video and optical flow. ↑ indicates larger values are better and vice versa. Training with *A2D* and *NP-Mix* yields considerable performance enhancements.

| Methods | HMDB51 25/26 | | | UCF101 50/51 | | | EPIC-Kitchens 4/4 | | | Kinetics-600 129/100 | | |
|---|---|---|---|---|---|---|---|---|---|---|---|---|
| | FPR95↓ | AUROC↑ | ID ACC↑ | FPR95↓ | AUROC↑ | ID ACC↑ | FPR95↓ | AUROC↑ | ID ACC↑ | FPR95↓ | AUROC↑ | ID ACC↑ |
| | *Without A2D Training* | | | | | | | | | | | |
| MSP | 44.66 | 87.74 | 89.32 | 22.14 | 95.73 | 99.22 | 76.31 | 67.59 | 71.46 | 64.08 | 76.16 | 80.11 |
| Energy | 43.36 | 87.46 | 89.32 | 22.52 | 96.06 | 99.22 | 76.68 | 68.29 | 71.46 | 68.75 | 75.49 | 80.11 |
| MaxLogit | 43.36 | 87.75 | 89.32 | 22.52 | 96.02 | 99.22 | 76.68 | 68.29 | 71.46 | 68.73 | 75.98 | 80.11 |
| Mahalanobis | 40.31 | 85.28 | 89.32 | 12.14 | 97.14 | 99.22 | 98.69 | 42.99 | 71.46 | 93.51 | 35.83 | 80.11 |
| ReAct | 42.05 | 87.79 | 89.32 | 25.63 | 95.85 | 99.32 | 83.96 | 65.89 | 71.08 | 72.40 | 73.80 | 80.35 |
| ASH | 53.59 | 87.16 | 89.54 | 32.14 | 94.02 | 99.22 | 76.87 | 67.92 | 70.15 | 69.24 | 76.16 | 79.62 |
| GEN | 43.79 | 87.49 | 89.32 | 23.79 | 95.54 | 99.22 | 76.87 | 68.52 | 71.46 | 69.03 | 75.33 | 80.11 |
| KNN | 42.92 | 88.46 | 89.32 | 15.63 | 96.93 | 99.22 | 75.93 | 63.60 | 71.46 | 68.67 | 74.64 | 80.11 |
| VIM | 36.82 | 88.06 | 89.32 | 12.52 | 97.66 | 99.22 | 77.05 | 65.60 | 71.46 | 68.77 | 75.47 | 80.11 |
| LogitNorm | 48.84 | 87.65 | 88.89 | 19.61 | 95.85 | 99.51 | 80.97 | 63.41 | 71.83 | 67.32 | 75.84 | 80.23 |
| | *With A2D Training* | | | | | | | | | | | |
| MSP+ | $38.78_{-5.88}$ | $88.37_{+0.63}$ | 90.63 | $7.09_{-15.05}$ | $98.19_{+2.46}$ | 99.61 | $66.23_{-10.08}$ | $71.04_{+3.45}$ | 71.46 | $63.04_{-1.04}$ | $76.47_{+0.31}$ | 79.50 |
| Energy+ | $39.22_{-4.14}$ | $88.84_{+1.38}$ | 90.63 | $9.81_{-12.71}$ | $98.16_{+2.10}$ | 99.61 | $66.98_{-9.70}$ | $72.45_{+4.16}$ | 71.46 | $64.59_{-4.16}$ | $76.45_{+0.96}$ | 79.50 |
| MaxLogit+ | $39.22_{-4.14}$ | $88.93_{+1.18}$ | 90.63 | $9.81_{-12.71}$ | $98.15_{+2.13}$ | 99.61 | $66.98_{-9.70}$ | $72.23_{+3.94}$ | 71.46 | $64.57_{-4.16}$ | $76.92_{+0.94}$ | 79.50 |
| Mahalanobis+ | $44.88_{+4.57}$ | $86.99_{+1.71}$ | 90.63 | $8.74_{-3.40}$ | $98.07_{+0.97}$ | 99.61 | $95.52_{-3.17}$ | $44.43_{+1.44}$ | 71.46 | $92.86_{-0.65}$ | $50.65_{+14.82}$ | 79.50 |
| ReAct+ | $37.91_{-4.14}$ | $89.09_{+1.30}$ | 90.63 | $10.39_{-15.24}$ | $98.12_{+2.27}$ | 99.61 | $68.66_{-15.30}$ | $72.03_{+6.14}$ | 70.52 | $71.26_{-1.14}$ | $74.29_{+0.49}$ | 79.64 |
| ASH+ | $42.05_{-11.54}$ | $87.72_{+0.56}$ | 90.41 | $12.52_{-19.62}$ | $97.43_{+3.41}$ | 99.42 | $63.25_{-13.62}$ | $74.73_{+6.81}$ | 67.72 | $64.28_{-4.96}$ | $77.28_{+1.12}$ | 79.15 |
| GEN+ | $38.56_{-5.23}$ | $88.61_{+1.12}$ | 90.63 | $8.83_{-14.96}$ | $98.12_{+2.58}$ | 99.61 | $65.86_{-11.01}$ | $73.05_{+4.53}$ | 71.46 | $62.28_{-6.75}$ | $77.08_{+1.75}$ | 79.50 |
| KNN+ | $33.33_{-9.59}$ | $89.59_{+1.13}$ | 90.63 | $7.48_{-8.15}$ | $98.39_{+1.46}$ | 99.61 | $72.39_{-3.54}$ | $67.83_{+4.23}$ | 71.46 | $65.71_{-2.96}$ | $76.23_{+1.59}$ | 79.50 |
| VIM+ | $33.77_{-3.05}$ | $89.37_{+1.31}$ | 90.63 | $6.80_{-5.72}$ | $98.62_{+0.96}$ | 99.61 | $66.42_{-10.63}$ | $68.43_{+2.83}$ | 71.46 | $64.59_{-4.18}$ | $76.45_{+0.98}$ | 79.50 |
| LogitNorm+ | $40.52_{-8.32}$ | $89.33_{+1.68}$ | 91.07 | $12.43_{-7.18}$ | $97.57_{+1.72}$ | 99.71 | $77.61_{-3.36}$ | $67.59_{+4.18}$ | 73.51 | $62.63_{-4.69}$ | $77.76_{+1.92}$ | 79.82 |
| | *With A2D Training and NP-Mix Outlier Synthesis* | | | | | | | | | | | |
| MSP++ | $33.99_{-10.67}$ | $88.79_{+1.05}$ | 89.98 | $7.96_{-14.18}$ | $98.24_{+2.51}$ | 99.71 | $67.16_{-9.15}$ | $71.52_{+3.93}$ | 71.64 | $62.91_{-1.17}$ | $76.92_{+0.76}$ | 80.52 |
| Energy++ | $36.38_{-6.98}$ | $88.91_{+1.45}$ | 89.98 | $6.50_{-16.02}$ | $98.48_{+2.42}$ | 99.71 | $67.91_{-8.77}$ | $73.79_{+5.50}$ | 71.64 | $63.69_{-5.06}$ | $77.11_{+1.62}$ | 80.52 |
| MaxLogit++ | $36.38_{-6.98}$ | $89.06_{+1.31}$ | 89.98 | $6.50_{-16.02}$ | $98.49_{+2.47}$ | 99.71 | $66.98_{-9.70}$ | $73.48_{+5.19}$ | 71.64 | $63.65_{-5.08}$ | $77.55_{+1.57}$ | 80.52 |
| Mahalanobis++ | $41.61_{+1.30}$ | $87.69_{+2.41}$ | 89.98 | $7.86_{-4.28}$ | $97.99_{+0.85}$ | 99.71 | $94.78_{-3.91}$ | $44.37_{+1.38}$ | 71.64 | $92.49_{-1.02}$ | $50.05_{+14.22}$ | 80.52 |
| ReAct++ | $37.47_{-4.58}$ | $88.63_{+0.84}$ | 90.20 | $12.43_{-13.20}$ | $97.33_{+1.48}$ | 99.90 | $66.60_{-17.36}$ | $73.11_{+7.22}$ | 72.39 | $67.95_{-4.45}$ | $75.55_{-1.75}$ | 80.70 |
| ASH++ | $36.17_{-17.42}$ | $89.30_{+2.14}$ | 89.32 | $10.68_{-21.46}$ | $97.80_{+3.78}$ | 99.81 | $66.23_{-10.64}$ | $73.06_{+5.14}$ | 66.23 | $63.77_{-5.47}$ | $77.44_{+1.28}$ | 79.44 |
| GEN++ | $35.95_{-7.84}$ | $89.78_{+2.29}$ | 89.98 | $7.67_{-16.12}$ | $98.28_{+2.74}$ | 99.71 | $65.30_{-11.57}$ | $75.17_{+6.65}$ | 71.64 | $62.95_{-6.08}$ | $76.95_{+1.62}$ | 80.52 |
| KNN++ | $33.77_{-9.15}$ | $90.05_{+1.59}$ | 89.98 | $12.04_{-3.59}$ | $97.65_{+0.72}$ | 99.71 | $71.27_{-4.66}$ | $69.94_{+6.34}$ | 71.64 | $66.81_{-1.86}$ | $74.19_{-0.45}$ | 80.52 |
| VIM++ | $34.64_{-2.18}$ | $88.80_{+0.74}$ | 89.98 | $4.56_{-7.96}$ | $98.73_{+1.07}$ | 99.71 | $67.72_{-9.33}$ | $69.09_{+3.49}$ | 71.64 | $63.67_{-5.10}$ | $77.12_{+1.65}$ | 80.52 |
| LogitNorm++ | $36.38_{-12.46}$ | $90.38_{+2.73}$ | 89.54 | $9.81_{-9.80}$ | $97.14_{+1.29}$ | 99.71 | $72.01_{-8.96}$ | $72.91_{+9.50}$ | 71.46 | $63.02_{-4.30}$ | $77.57_{+1.73}$ | 80.33 |

where $\lambda \sim \text{Beta}(\alpha, \alpha)$, for $\alpha \in (0, \infty)$. In our experiments, we opt for $\alpha > 1$ to ensure the synthesized outliers reside in the intermediary space between two prototypes, rather than near the ID data. We then partition $\widetilde{\mathbf{Z}}$ into different modalities as $\widetilde{\mathbf{Z}} = [\widetilde{\mathbf{Z}}^1, \widetilde{\mathbf{Z}}^2]$, where $\widetilde{\mathbf{Z}}^1$ and $\widetilde{\mathbf{Z}}^2$ represent the synthesized outlier embeddings for each modality. Subsequently, the corresponding prediction probabilities are computed as $\widetilde{p}^1 = \delta(h_1(\widetilde{\mathbf{Z}}^1))$ and $\widetilde{p}^2 = \delta(h_2(\widetilde{\mathbf{Z}}^2))$.

For synthesized outliers, we aim to maximize the prediction discrepancies between different modalities, similar to Eq. (2) for ID training samples. In this context, there is no need to remove any value from the prediction, as the outliers are assumed not to belong to any ID class. The discrepancy loss between $\widetilde{p}^1$ and $\widetilde{p}^2$ can be defined as:

$$\mathcal{L}_{Discr\_syn} = -Discr(\widetilde{p}^1, \widetilde{p}^2). \tag{7}$$

Moreover, we seek to mitigate the overconfidence of predictions for synthesized outliers towards any existing ID class. Therefore, we maximize the entropy of predictions:

$$\mathcal{L}_{Ent} = -\frac{1}{2}(H(\widetilde{p}^1) + H(\widetilde{p}^2)), \tag{8}$$

where $H(\widetilde{p})$ is the entropy of prediction $\widetilde{p}$ and can be calculated as $H(\widetilde{p}) = -\sum_c \widetilde{p}_c \log \widetilde{p}_c$. The final loss is obtained as the weighted sum of the previously defined losses:

$$\mathcal{L} = \mathcal{L}_{cls} + \gamma(\mathcal{L}_{Discr} + \mathcal{L}_{Discr\_syn} + \mathcal{L}_{Ent}). \tag{9}$$

## 5 Experiments

### 5.1 Experimental Setting

**Baselines.** Our objective is to enhance existing unimodal OOD algorithms through multimodal training utilizing the proposed *A2D* and *NP-Mix*. To validate the efficacy and versatility of our proposed approaches, we include representative algorithms that leverage information from the probability space (MSP [31], GEN [44]), logit space (Energy [43], MaxLogit [30]), feature space (Mahalanobis [41], KNN [59]), both logit and feature spaces (VIM [62]), penultimate activation manipulations (ReAct [58], ASH [16]), and training-time regularization (LogitNorm [65]).

**Evaluation metrics.** We evaluate the performance via the use of the following metrics: (1) the false positive rate (FPR95) of OOD samples when the true positive rate of ID samples is at 95%, (2) the

Table 2: **Multimodal Far-OOD Detection** using video and optical flow, with **HMDB51** as ID.

| Methods | OOD Datasets | | | | | | | | | | ID ACC ↑ |
|---|---|---|---|---|---|---|---|---|---|---|---|
| | Kinetics-600 | | UCF101 | | EPIC-Kitchens | | HAC | | Average | | |
| | FPR95↓ | AUROC↑ | FPR95↓ | AUROC↑ | FPR95↓ | AUROC↑ | FPR95↓ | AUROC↑ | FPR95↓ | AUROC↑ | |
| | **Without A2D Training** | | | | | | | | | | |
| Energy | 32.95 | 92.48 | 44.93 | 87.95 | 8.10 | 97.70 | 32.95 | 92.28 | 29.73 | 92.60 | 87.23 |
| ASH | 51.20 | 87.81 | 53.93 | 84.22 | 19.95 | 95.92 | 42.99 | 90.23 | 42.02 | 89.55 | 86.20 |
| GEN | 41.51 | 90.34 | 46.18 | 87.91 | 8.21 | 98.26 | 38.31 | 91.28 | 33.55 | 91.95 | 87.23 |
| KNN | 22.69 | 95.01 | 39.34 | 89.28 | 9.92 | 97.92 | 20.75 | 96.02 | 23.18 | 94.56 | 87.23 |
| VIM | 13.68 | 97.01 | 33.87 | 91.45 | 5.93 | 98.15 | 13.45 | 97.12 | 16.73 | 95.93 | 87.23 |
| | **With A2D Training and NP-Mix Outlier Synthesis** | | | | | | | | | | |
| Energy++ | $24.52_{-8.43}$ | $93.96_{+1.48}$ | $36.49_{-8.44}$ | $89.67_{+1.72}$ | $6.96_{-1.14}$ | $97.53_{-0.17}$ | $22.92_{-10.14}$ | $94.41_{+2.13}$ | $22.72_{-7.01}$ | $93.89_{+1.29}$ | 86.89 |
| ASH++ | $27.82_{-23.38}$ | $93.17_{+5.36}$ | $38.43_{-15.50}$ | $89.52_{+5.30}$ | $6.84_{-13.11}$ | $98.23_{+2.31}$ | $23.03_{-19.96}$ | $94.45_{+4.22}$ | $24.03_{-17.99}$ | $93.84_{+4.29}$ | 86.20 |
| GEN++ | $25.66_{-15.85}$ | $93.50_{+3.16}$ | $37.40_{-8.78}$ | $91.19_{+3.28}$ | $5.25_{-2.96}$ | $98.98_{+0.72}$ | $24.63_{-13.68}$ | $94.28_{+3.00}$ | $23.24_{-10.31}$ | $94.49_{+2.54}$ | 86.89 |
| KNN++ | $15.05_{-7.64}$ | $96.96_{+1.95}$ | $33.06_{-6.28}$ | $91.92_{+2.64}$ | $5.47_{-4.45}$ | $98.97_{+1.05}$ | $13.45_{-7.30}$ | $97.25_{+1.23}$ | $16.76_{-6.42}$ | $96.28_{+1.72}$ | 86.89 |
| VIM++ | $9.24_{-4.44}$ | $98.04_{+1.03}$ | $26.45_{-7.42}$ | $92.34_{+0.89}$ | $5.36_{-0.57}$ | $98.09_{-0.06}$ | $6.04_{-7.41}$ | $98.56_{+1.44}$ | $11.77_{-4.96}$ | $96.76_{+0.83}$ | 86.89 |

Table 3: **Multimodal Far-OOD Detection** using video and optical flow, with **Kinetics-600** as ID.

| Methods | OOD Datasets | | | | | | | | | | ID ACC ↑ |
|---|---|---|---|---|---|---|---|---|---|---|---|
| | HMDB51 | | UCF101 | | EPIC-Kitchens | | HAC | | Average | | |
| | FPR95↓ | AUROC↑ | FPR95↓ | AUROC↑ | FPR95↓ | AUROC↑ | FPR95↓ | AUROC↑ | FPR95↓ | AUROC↑ | |
| | **Without A2D Training** | | | | | | | | | | |
| Energy | 72.64 | 71.75 | 70.12 | 71.49 | 43.66 | 82.05 | 61.50 | 74.99 | 61.98 | 75.07 | 73.14 |
| ASH | 71.62 | 76.66 | 69.36 | 72.38 | 34.38 | 88.05 | 47.85 | 83.49 | 55.80 | 80.15 | 72.20 |
| GEN | 68.47 | 78.43 | 64.80 | 73.97 | 36.81 | 85.11 | 49.53 | 83.67 | 54.90 | 80.30 | 73.14 |
| KNN | 71.08 | 78.84 | 68.62 | 74.33 | 41.83 | 82.32 | 57.00 | 82.53 | 59.63 | 79.51 | 73.14 |
| VIM | 72.25 | 71.88 | 70.72 | 70.58 | 43.14 | 82.69 | 59.48 | 75.46 | 61.40 | 75.15 | 73.14 |
| | **With A2D Training and NP-Mix Outlier Synthesis** | | | | | | | | | | |
| Energy++ | $63.27_{-9.37}$ | $74.17_{+2.42}$ | $67.20_{-2.92}$ | $74.50_{+3.01}$ | $34.07_{-9.59}$ | $87.49_{+5.44}$ | $56.69_{-4.81}$ | $80.20_{+5.21}$ | $55.31_{-6.67}$ | $79.09_{+4.02}$ | 73.67 |
| ASH++ | $63.65_{-7.97}$ | $79.14_{+2.48}$ | $62.15_{-7.21}$ | $75.32_{+2.94}$ | $36.09_{+1.71}$ | $88.69_{+0.64}$ | $46.47_{-1.38}$ | $85.21_{+1.72}$ | $52.09_{-3.71}$ | $82.09_{+1.94}$ | 72.53 |
| GEN++ | $61.67_{-6.80}$ | $80.63_{+2.20}$ | $57.03_{-7.77}$ | $77.97_{+4.00}$ | $40.09_{+3.28}$ | $85.10_{-0.01}$ | $41.97_{-7.56}$ | $86.98_{+3.31}$ | $50.19_{-4.71}$ | $82.67_{+2.37}$ | 73.67 |
| KNN++ | $64.60_{-6.48}$ | $80.28_{+1.44}$ | $67.44_{-1.18}$ | $77.95_{+3.62}$ | $37.65_{-4.18}$ | $83.55_{+1.23}$ | $53.63_{-3.37}$ | $83.34_{+0.81}$ | $55.83_{-3.80}$ | $81.28_{+1.77}$ | 73.67 |
| VIM++ | $63.25_{-9.00}$ | $73.96_{+2.08}$ | $66.31_{-4.41}$ | $74.08_{+3.50}$ | $34.45_{-8.69}$ | $87.67_{+4.98}$ | $53.82_{-5.66}$ | $81.06_{+5.60}$ | $54.46_{-6.94}$ | $79.19_{+4.04}$ | 73.67 |

area under the receiver operating characteristic curve (AUROC), and (3) ID classification accuracy (ID ACC). For each baseline algorithm, we report the results both with and without *A2D* training and *NP-Mix* outlier synthesis. Additional implementation details are provided in Appendix D.

## 5.2 Multimodal Near-OOD Detection

We commence our evaluation by assessing the efficacy of *A2D* training and *NP-Mix* outlier synthesis within the Multimodal Near-OOD Detection setup. As presented in Tab. 1, the simple incorporation of *A2D* yields substantial enhancements in OOD performance across nearly all scenarios. The average prediction $L_1$ distances between ID and OOD data ($l_{OOD} - l_{ID}$) before and after *A2D* training across various datasets are depicted in Fig. 4. Notably, across all datasets, *A2D* training serves to further enlarge the Modality Prediction Discrepancy and consequently improve OOD detection performance, verifying the strong correlation between them. Specifically, *A2D* training reduces the FPR95 by up to absolute $19.62\%$ on UCF101 50/51 dataset for ASH and increases AUROC up to $14.82\%$ for Mahalanobis on Kinetics-600 129/100. Combining *A2D* training with *NP-Mix* yields additional improvements in OOD detection performance, underscoring the efficacy of outlier synthesis in model regularization. Additionally, *A2D* training and *NP-Mix* outlier synthesis exhibit notable efficacy across all baseline OOD detection algorithms despite their different underlying principles, indicating the versatility of our proposed method.

## 5.3 Multimodal Far-OOD Detection

Tab. 2 and Tab. 3 present the results of Multimodal Far-OOD Detection with HMDB51 and Kinetics-600 as ID datasets, respectively. Similar to the Near-OOD setup, training with *A2D* and *NP-Mix* yields considerable enhancements in OOD detection performance across most cases for all baseline algorithms. Specifically, training with both *A2D* and *NP-Mix* achieves reductions in FPR95 of up to absolute $23.38\%$ on the HMDB51 dataset for ASH and up to $14.43\%$ for ReAct on the Kinetics-600 dataset. Due to space limits, we provide comprehensive results in Appendix E (Tab. 7 and Tab. 8). Interestingly, we observe that the performance on the HMDB51 dataset generally surpasses that of the Kinetics-600 dataset. This finding aligns with observations from unimodal OOD detection benchmarks [66], where performance on datasets with fewer classes (e.g., CIFAR-10 [38]) tends to outperform those on datasets with a greater number of classes (e.g., ImageNet [14]).

Table 4: **Multimodal Near-OOD Detection** using video, audio, and optical flow.

| Methods | Modality | | | EPIC-Kitchens 4/4 | | | Kinetics-600 129/100 | | |
|---|---|---|---|---|---|---|---|---|---|
| | Video | Audio | Flow | FPR95↓ | AUROC↑ | ID ACC↑ | FPR95↓ | AUROC↑ | ID ACC↑ |
| | | | | **Without A2D Training** | | | | | |
| Energy | ✓ | ✓ | ✓ | 69.22 | 72.39 | 73.13 | 66.42 | 76.60 | 80.33 |
| ASH | ✓ | ✓ | ✓ | 70.52 | 69.70 | 68.10 | 63.48 | 78.11 | 79.54 |
| GEN | ✓ | ✓ | ✓ | 70.34 | 70.99 | 73.13 | 64.24 | 77.54 | 80.33 |
| KNN | ✓ | ✓ | ✓ | 80.22 | 60.56 | 73.13 | 75.03 | 65.97 | 80.33 |
| VIM | ✓ | ✓ | ✓ | 76.12 | 59.03 | 73.13 | 66.38 | 76.59 | 80.33 |
| | | | | **With A2D Training and NP-Mix Outlier Synthesis** | | | | | |
| Energy++ | ✓ | ✓ | ✓ | $62.69_{-6.53}$ | $74.95_{+2.56}$ | 71.46 | $63.81_{-2.61}$ | $77.89_{+1.29}$ | 80.82 |
| ASH++ | ✓ | ✓ | ✓ | $69.78_{-0.74}$ | $69.49_{-0.21}$ | 67.72 | $61.22_{-2.26}$ | $78.57_{+0.46}$ | 80.05 |
| GEN++ | ✓ | ✓ | ✓ | $63.62_{-6.72}$ | $73.94_{+2.95}$ | 71.46 | $63.55_{-0.69}$ | $77.92_{+0.38}$ | 80.82 |
| KNN++ | ✓ | ✓ | ✓ | $68.47_{-11.75}$ | $67.87_{+7.31}$ | 71.46 | $71.46_{-3.57}$ | $68.87_{+2.90}$ | 80.82 |
| VIM++ | ✓ | ✓ | ✓ | $73.51_{-2.61}$ | $59.57_{+0.54}$ | 71.46 | $63.42_{-2.96}$ | $77.90_{+1.31}$ | 80.82 |

Table 5: Ablation on **Distance Functions** for Near-OOD Detection on HMDB51 dataset.

| Methods | Baseline | | $L_1$ | | $L_2$ | | Wasserstein | | Hellinger | |
|---|---|---|---|---|---|---|---|---|---|---|
| | FPR95↓ | AUROC↑ | FPR95↓ | AUROC↑ | FPR95↓ | AUROC↑ | FPR95↓ | AUROC↑ | FPR95↓ | AUROC↑ |
| Energy | 43.36 | 87.46 | 37.04 | 88.52 | 37.69 | 88.84 | 36.17 | 88.61 | 36.38 | 88.91 |
| GEN | 43.79 | 87.49 | 39.87 | 90.34 | 42.27 | 88.84 | 37.91 | 88.76 | 35.95 | 89.78 |
| KNN | 42.92 | 88.46 | 35.73 | 90.24 | 35.51 | 89.78 | 36.60 | 88.57 | 33.77 | 90.05 |
| VIM | 36.82 | 88.06 | 32.24 | 89.36 | 33.99 | 89.03 | 33.99 | 89.24 | 34.64 | 88.80 |

Table 6: Ablation on **Outlier Synthesis Methods** for Near-OOD Detection on HMDB51 dataset.

| Methods | Baseline | | VOS | | NPOS | | Mixup | | NP-Mix | |
|---|---|---|---|---|---|---|---|---|---|---|
| | FPR95↓ | AUROC↑ | FPR95↓ | AUROC↑ | FPR95↓ | AUROC↑ | FPR95↓ | AUROC↑ | FPR95↓ | AUROC↑ |
| Energy | 43.36 | 87.46 | 37.69 | 87.94 | 43.14 | 87.59 | 42.27 | 87.77 | 36.38 | 88.91 |
| GEN | 43.79 | 87.49 | 37.69 | 89.51 | 44.44 | 88.11 | 42.70 | 88.28 | 35.95 | 89.78 |
| KNN | 42.92 | 88.46 | 38.34 | 88.55 | 40.52 | 88.81 | 36.17 | 89.61 | 33.77 | 90.05 |
| VIM | 36.82 | 88.06 | 35.51 | 88.93 | 37.69 | 88.17 | 37.04 | 88.44 | 34.64 | 88.80 |

## 5.4 Ablation Studies

**Multimodal Near-OOD Detection with other modalities.** We demonstrate the efficacy of *A2D* training and *NP-Mix* outlier synthesis across various combinations of modalities, not limited to video and optical flow, as detailed in Tab. 4 and Tab. 9 in Appendix E. Notably, the performance of most algorithms is consistently improved with *A2D* and *NP-Mix*, regardless of which combinations of modalities are used.

**Effectiveness of other distance functions.** We substitute the distance metric for measuring probability distributions in the discrepancy loss with alternative distance functions, including $L_1$, $L_2$, and Wasserstein [2] distances. As depicted in Tab. 5, *A2D* training exhibits robustness across various distance functions. Regardless of the specific distance metric employed, substantial improvements are consistently observed compared to the baseline approach without *A2D* training.

**Compared with other outlier synthesis methods.** To evaluate the effectiveness of other outlier synthesis methods, we replace *NP-Mix* with VOS [22], NPOS [60], and Mixup [69] and train with *A2D*. All baseline methods yield improvements in OOD performance, underscoring the significance of outlier synthesis for model regularization. Notably, *NP-Mix* emerges as the most effective among the various baselines by synthesizing outliers that span broader embedding spaces.

## 6 Conclusion

In this paper, we introduce the Multimodal Out-of-distribution Detection problem and present a novel benchmark, MultiOOD, which includes diverse dataset sizes and various combinations of modalities. Motivated by the *Modality Prediction Discrepancy* phenomenon observed within MultiOOD, we propose a novel *A2D* training algorithm to encourage the enlargement of such discrepancy during model training, along with a new outlier synthesis algorithm, *NP-Mix*, that complements *A2D*. Extensive experiments on MultiOOD and under Near-OOD and Far-OOD setups verify the efficacy of the proposed approaches. We hope our work will inspire future research endeavors in Multimodal OOD Detection.

**Limitation and Future Work.** The performance on Near-OOD benchmarks and on datasets with a large number of classes still exhibits potential for enhancement. Moreover, the exploration of Outlier Exposure [32] deserves attention as a potential to better learn ID/OOD discrepancy.

## Acknowledgments

The authors acknowledge the support of "In-service diagnostics of the catenary/pantograph and wheelset axle systems through intelligent algorithms" (SENTINEL) project, supported by the ETH Mobility Initiative.

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

# A  Related Work

**Out-of-Distribution (OOD) Detection** aims to detect test samples with semantic shift without losing the ID classification accuracy. Numerous OOD detection algorithms have been developed, with post hoc methods and training-time regularization as major categories [66]. *Post hoc* methods aim to design OOD scores based on the classification output of neural networks, offering the advantage of being easy to use without modifying the training procedure and objective. Early approaches include utilizing Maximum Softmax Probability (MSP) [31] as OOD score, often coupled with temperature scaling and input perturbation [42]. Instead of using softmax probabilities, MaxLogit [30] employs maximum logit as OOD scores rather than softmax. Energy-based algorithm [43] demonstrated the efficacy of energy function [40] in quantifying OOD-ness. Other approaches like ReAct [58] improved existing scoring functions by truncating the activations with high values. Similarly, ASH [16] prunes a large portion of the input activations and adjusts the remaining activations using pruning, binarizing, or scaling. Methods like Mahalanobis [41] and $k$-Nearest Neighbor (KNN) [59] use distance metrics in feature space for OOD detection, while Virtual-logit Matching (VIM) [62] integrates information from both feature and logit spaces to define the OOD score. Recently, Generalized Entropy (GEN) [44] proposed an entropy-based score function that proves to be both simple and effective.

*Training-time regularization* methods such as LogitNorm [65] address prediction overconfidence by imposing a constant vector norm on the logits during training. Outlier Exposure [32] leverages external OOD samples from other datasets during training to facilitate the learning of better ID and OOD discrepancy. Additionally, some approaches [22, 60] proposed synthesizing virtual outliers for training-time regularization. However, all previous approaches were designed for unimodal scenarios, without accounting for the interaction and complementary nature of diverse modalities.

**Multimodal OOD Detection.** Recent endeavors [47, 63] have explored vision-language models [53] to enhance OOD performance, which are also referred to as *multimodal* in some of the studies. Maximum Concept Matching (MCM) [47] defines OOD score by aligning visual features with textual concepts. CLIPN [63] equips CLIP [53] with the capability of distinguishing OOD and ID samples using positive-semantic prompts and negation-semantic prompts. However, the evaluations of all these works are still limited to benchmarks *exclusively containing images*. Consequently, existing methods fall short in fully leveraging the complementary information from various modalities, such as LiDAR and camera in autonomous driving [21], as well as video, audio, and optical flow in action recognition [56]. There is also a lack of benchmark datasets that facilitate the evaluation of Multimodal OOD Detection. Therefore, we aim to develop a more practical and challenging benchmark incorporating multiple combinations of modalities (i.e., video, audio, and optical flow). This enables the creation of OOD detection algorithms that are specifically designed to leverage the complementary nature of various modalities effectively.

**Anomaly Detection and Open Set Recognition** are two closely related fields to OOD Detection. *Anomaly Detection* (AD) aims to detect patterns that deviate from the predefined normality during testing [67] and treats all in-distribution samples as a single class. Therefore, AD algorithms can be applied to OOD detection by ignoring all the labels for ID data. Typical AD algorithms include unsupervised [54, 5], semi-supervised [55, 71], and supervised [26, 52], depending on the availability of labels [28]. *Open Set Recognition* (OSR) [61, 37, 9, 17] focuses on accurately classifying test samples from "known known classes" (ID) and detecting test samples from "unknown unknown classes" (OOD). While OOD detection benchmarks always take one dataset as ID and find several other datasets with non-overlapping categories as OOD, OSR benchmarks usually split one multi-class classification dataset into ID and OOD parts according to classes. However, both OSR and OOD detection share the same goal of identifying test samples with *semantic shifts* without compromising the accuracy of ID classification [67]. Therefore, in our benchmark, we treat OSR and OOD as synonymous concepts and adopt OOD as the general term. Our *Near-OOD* Benchmark is similar to traditional OSR setup and our *Far-OOD* Benchmark is the same as general OOD setup.

**OOD Benchmarks.** Early works [31] in OOD detection primarily focus on small-scale image datasets such as MNIST [15] and CIFAR-10/100 [38]. Recognizing the need for evaluating OOD detection at scale, studies such as [62] introduce new OOD datasets based on ImageNet [14]. Additionally, some OOD benchmarks [30, 8] are specifically designed for semantic segmentation tasks. OpenOOD [66] offers a comprehensive OOD benchmark comprising datasets from previous works and incorporating over 30 common OOD methods. However, all of these benchmarks are limited to *image* data. In contrast, our MultiOOD is the first public OOD benchmark that encompasses different combinations of modalities, facilitating future research endeavors in Multimodal OOD Detection.

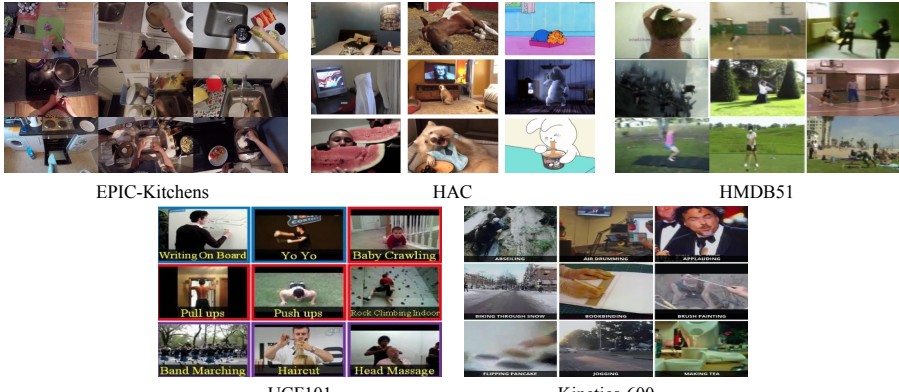

Figure 6: Visualization of action recognition datasets used in our MultiOOD benchmark.

# B More Details on the MultiOOD Benchmark

## B.1 Datasets Used in MultiOOD Benchmark

Action recognition is inherently multimodal and serves as the primary task within our benchmark, and we include five action recognition datasets accordingly of varying sizes, as shown in Fig. 6.

**EPIC-Kitchens [13].** The EPIC-Kitchens dataset is a large-scale egocentric dataset collected by 32 participants in their native kitchen environments. The participants were asked to capture all their daily kitchen activities and record sequences regardless of their duration. The start and end times for each action are annotated. We use a subset of the EPIC-Kitchens dataset introduced in the Multimodal Domain Adaptation paper [48], which comprises 4, 871 video clips from 8 largest action classes in sequence P22. These actions include 'put', 'take', 'open', 'close', 'wash', 'cut', 'mix', and 'pour', with provided modalities including *video*, *optical flow*, and *audio*.

**HAC [20].** The HAC dataset encompasses seven actions ('sleeping', 'watching tv', 'eating', 'drinking', 'swimming', 'running', and 'opening door') performed by humans, animals, and cartoon figures. There are 3, 381 video clips in total from seven actions. Modalities provided in this dataset include *video*, *optical flow*, and *audio*.

**HMDB51 [39].** The HMDB51 dataset is a video action recognition dataset, comprising 6, 766 video clips spanning 51 action categories. The video clips are extracted from a variety of sources ranging from digitized movies to YouTube. Available modalities in this dataset include *video* and *optical flow*.

**UCF101 [57].** UCF101 is another video action recognition dataset collected from YouTube, comprising 13, 320 video clips from 101 actions. UCF101 offers substantial diversity in action types and encompasses significant variations in camera motion, object appearance and pose, object scale, viewpoint, cluttered background, illumination conditions, etc. Modalities provided in this dataset include *video* and *optical flow*.

**Kinetics-600 [6].** Kinetics-600 is a large-scale action recognition dataset, featuring approximately 480$k$ videos spanning 600 action categories. Each video in the dataset is a 10-second clip of action moment annotated from YouTube videos. In our benchmark, we carefully selected a subset of 229 action classes from Kinetics-600 to mitigate the potential category overlap with other datasets, with each class comprising roughly 250 video clips, yielding a total of 57, 205 video clips. The original dataset provides *video* and *audio* modalities. To make it consistent with the other dataset, our benchmark further provides the extracted *optical flow* for all video clips, amounting to a total of 114, 410 optical flow samples. The dense optical flow is extracted at 24 frames per second using the TV-L1 algorithm [68].

## B.2 Multimodal Near-OOD Benchmark

In the *Near-OOD* setup, we incorporate four datasets. **EPIC-Kitchens 4/4** is derived from the EPIC-Kitchens Domain Adaptation dataset [48], where the dataset is randomly partitioned into four classes for training as ID and four classes for testing as OOD. Similarly, **HMDB51 25/26** and **UCF101 50/51**

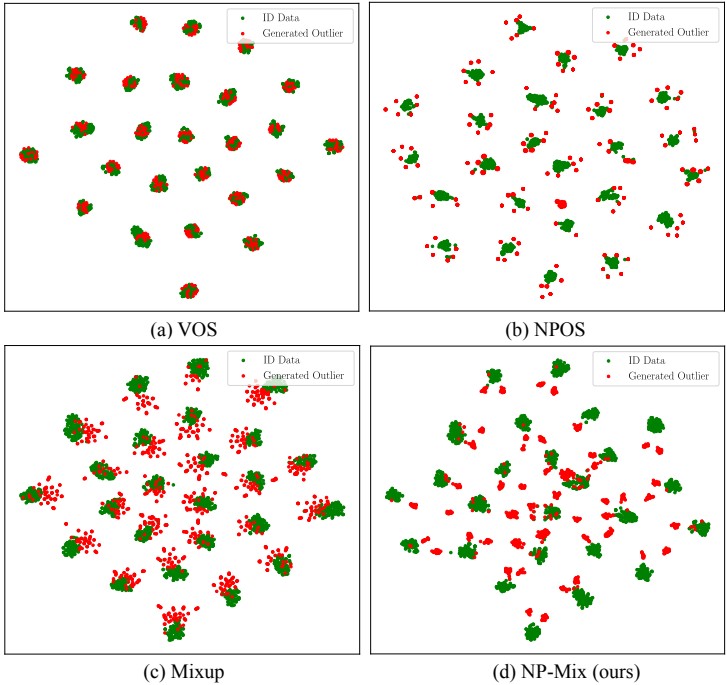

|  |  |
|---|---|
| (a) VOS | (b) NPOS |
| (c) Mixup | (d) NP-Mix (ours) |

Figure 7: Visualization of synthesized outliers compared against other methods. VOS and NPOS tend to generate outliers near the ID data, neglecting to explore the broader embedding space. Mixup randomly selects samples from all classes to mix and inadvertently introduces unwanted noise samples within the distribution of ID data. *NP-Mix* excels at generating synthesized outliers by effectively utilizing information from neighbor classes and spanning wider embedding spaces.

are constructed based on HMDB51 [39] and UCF101 [57], respectively. In the case of **Kinetics-600 129/100**, we curate 229 action classes from the Kinetics-600 dataset [6], with each class comprising approximately 250 video clips. Within this setup, 129 classes are randomly designated for training as ID, while the remaining 100 classes are allocated for testing as OOD.

## B.3 Multimodal Far-OOD Benchmark

In the *Far-OOD* setup, we incorporate HMDB51 and Kinetics-600 as ID datasets.

**HMDB51 dataset as ID.** For the OOD datasets, we utilize UCF101, EPIC-Kitchens, HAC, and Kinetics-600 datasets. All of these datasets are carefully curated to remove samples that should belong to ID classes. Given the relatively small number of classes in the EPIC-Kitchens and HAC datasets, we remove 8 classes in the HMDB51 dataset that overlap with EPIC-Kitchens and HAC, including 'chew', 'climb_stairs', 'drink', 'eat', 'pick', 'pour', 'ride_horse', 'run', leaving 43 classes as ID classes. In the case of UCF101, we remove 31 overlapping classes with HMDB51, resulting in 70 classes designated as OOD classes for evaluation. For Kinetics-600, we use the same subset of 229 classes as in the Near-OOD setup, which are carefully selected to mitigate the potential category overlap with other datasets. For other datasets, no class overlap exists and we utilize their original classes as OOD.

**Kinetics-600 dataset as ID.** Similarly, we adopt UCF101, EPIC-Kitchens, HAC, and HMDB51 datasets as OOD datasets, with careful curation undertaken to exclude samples belonging to ID classes. We carefully selected a subset of 229 action classes from Kinetics-600 in the *Near-OOD* setup to mitigate the potential overlap with other datasets. For the UCF101 dataset, we remove 11 overlapping classes with Kinetics-600, leaving 90 classes as OOD classes for evaluation. For other datasets, there are no class overlap issues, and we use their original classes as OOD.

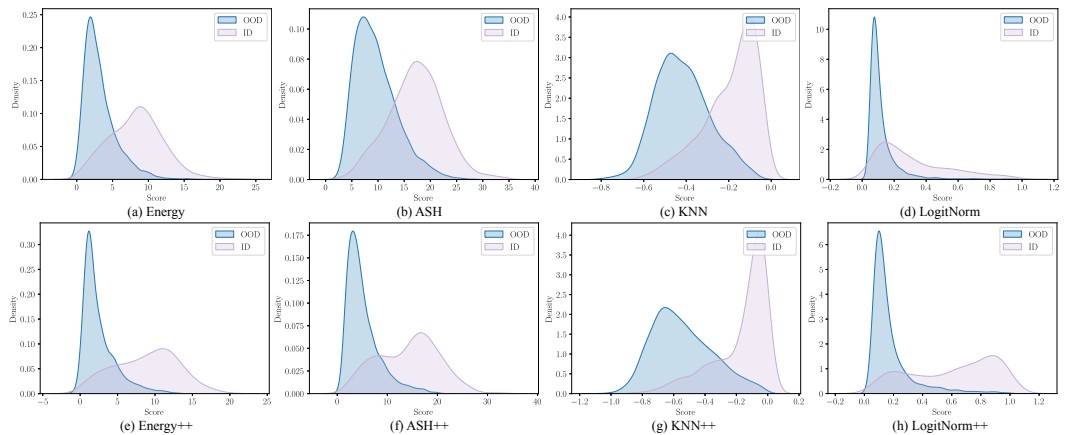

Figure 8: Score distributions of different baseline methods on the HMDB51 25/26 dataset before and after training with *A2D* and *NP-Mix*.

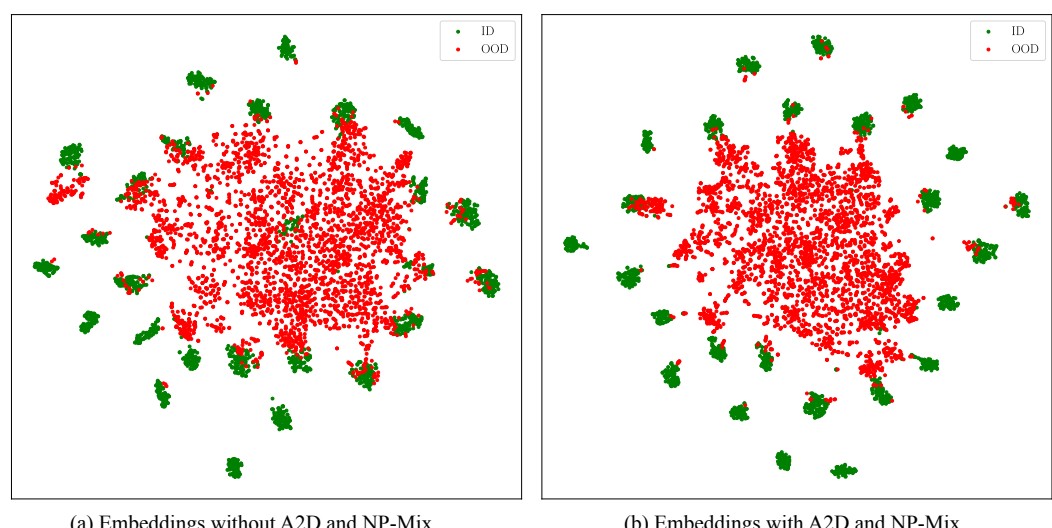

(a) Embeddings without A2D and NP-Mix       (b) Embeddings with A2D and NP-Mix

Figure 9: Visualization of the learned embeddings on ID and OOD data using t-SNE on the HMDB51 25/26 dataset before and after training with *A2D* and *NP-Mix*.

## C Visualization of Results

**Visualization of Synthesized Outliers.** We visualize the outliers generated by different outlier synthesis algorithms, including VOS [22], NPOS [60], Mixup [69], and our proposed *NP-Mix*. As shown in Fig. 7, VOS and NPOS generate outliers near the ID data, neglecting to explore the broader embedding space. Mixup randomly selects samples from all classes to mix and introduces unwanted noise samples within the distribution of ID data. *NP-Mix* excels at generating synthesized outliers by effectively utilizing information from neighboring classes and spanning wider embedding spaces.

**Score Distributions.** Fig. 8 illustrates the score distributions generated by various baseline methods on the HMDB51 25/26 dataset before and after training with *A2D* and *NP-Mix*. Score distributions produced by *A2D* and *NP-Mix* lead to better ID/OOD separation, resulting in strengthened OOD detection performance.

**Visualization of Learned Embeddings for ID and OOD Data.** Fig. 9 shows the visualization of the learned embeddings using t-SNE [46] on the HMDB51 25/26 dataset before and after training with *A2D* and *NP-Mix*. The embedding of ID and OOD data are more separable after *A2D* training and *NP-Mix* outlier synthesis.

## D More Implementation Details

### D.1 Training Details

We conduct experiments across three modalities: video, audio, and optical flow. We adopt the MMAction2 [11] toolkit for experiments. To encode the visual information, we utilize SlowFast network [23], initialized with pre-trained weights from Kinetics-400 [34]. For the audio encoder, we employ ResNet-18 [29] , initializing the weights from the VGGSound pre-trained checkpoint [10]. Similarly, we use the SlowFast network with a slow-only pathway, again leveraging pre-trained weights from Kinetics-400 [34] for the optical flow encoder. We use the Adam optimizer [36] with a learning rate of 0.0001 and a batch size of 16. Additionally, we set the hyperparameters as follows: $\gamma = 0.5$, mixup $\alpha = 10.0$, nearest neighbor $N = 2$. We train the network for 50 epochs on an RTX 3090 GPU and select the model with the best performance on the validation dataset.

### D.2 Extension to More Modalities

Our framework is not limited to two modalities and can be easily extended to $M$ modalities. Given a training sample $\mathbf{x}$ with label $c$ and $M$ modalities, we obtain prediction probabilities $\hat{p}$ from the combined embeddings of all modalities, and $\hat{p}^1, \hat{p}^2, ..., \hat{p}^M$ from each modality, all of which are of shape $[1, C]$, where $C$ represents the number of classes. By removing the $c$-th value from each prediction, we derive new prediction probabilities without ground-truth classes, denoted as $\bar{p}^1, \bar{p}^2, ..., \bar{p}^M$ with shapes $[1, C-1]$. Subsequently, we aim to maximize the discrepancy between $\bar{p}^1, \bar{p}^2, ..., \bar{p}^M$, which can be defined as:

$$\mathcal{L}_{Discr} = -\frac{2}{M(M-1)} \sum_{i=1}^{M-1} \sum_{j=i+1}^{M} Discr(\bar{p}^i, \bar{p}^j). \tag{10}$$

where $Discr(\cdot)$ is a distance metric quantifying the similarity between two probability distributions. Similarly, for $\mathcal{L}_{cls}$, $\mathcal{L}_{Discr\_syn}$ and $\mathcal{L}_{Ent}$, we can define as:

$$\mathcal{L}_{cls} = \frac{1}{M+1}(CE(\hat{p}, y) + \sum_{i=1}^{M} CE(\hat{p}^i, y)), \tag{11}$$

$$\mathcal{L}_{Discr\_syn} = -\frac{2}{M(M-1)} \sum_{i=1}^{M-1} \sum_{j=i+1}^{M} Discr(\widetilde{p}^i, \widetilde{p}^j), \tag{12}$$

$$\mathcal{L}_{Ent} = -\frac{1}{M} \sum_{i=1}^{M} H(\widetilde{p}^i). \tag{13}$$

The final loss is obtained as the weighted sum of the previously defined losses:

$$\mathcal{L} = \mathcal{L}_{cls} + \gamma(\mathcal{L}_{Discr} + \mathcal{L}_{Discr\_syn} + \mathcal{L}_{Ent}). \tag{14}$$

### D.3 Inference Details

For algorithms that define OOD score or truncate activations leveraging information from the feature space (Mahalanobis [41], KNN [59], VIM [62], ReAct [58], ASH [16]), we use the combined embedding $\mathbf{Z} = [\mathbf{Z}^1, \mathbf{Z}^2, ..., \mathbf{Z}^M]$ from all modalities. For algorithms that define the OOD score leveraging information from the probability space or logit space (MSP [31], GEN [44], Energy [43], MaxLogit [30], VIM [62], LogitNorm [65]), we use the prediction probabilities $\hat{p}$ or prediction logits $h(\mathbf{Z})$ obtained from the combined embeddings of all modalities.

## E Further Experimental Results

**Multimodal Far-OOD Detection.** Tab. 7 and Tab. 8 present comprehensive results for Multimodal Far-OOD Detection on the HMDB51 and Kinetics-600 datasets. Training with *A2D* and *NP-Mix* significantly improves OOD performance in most cases across all baseline algorithms, underscoring the versatility of our proposed method.

Table 7: **Multimodal Far-OOD Detection** using video and optical flow, with **HMDB51** as ID.

| Methods | Kinetics-600 | | UCF101 | | EPIC-Kitchens | | HAC | | Average | | ID ACC ↑ |
|---|---|---|---|---|---|---|---|---|---|---|---|
| | FPR95↓ | AUROC↑ | FPR95↓ | AUROC↑ | FPR95↓ | AUROC↑ | FPR95↓ | AUROC↑ | FPR95↓ | AUROC↑ | |
| | | | | | | Without A2D Training | | | | | |
| MSP | 39.11 | 88.78 | 46.64 | 86.40 | 17.33 | 95.99 | 39.91 | 89.10 | 35.75 | 90.07 | 87.23 |
| Energy | 32.95 | 92.48 | 44.93 | 87.95 | 8.10 | 97.70 | 32.95 | 92.28 | 29.73 | 92.60 | 87.23 |
| MaxLogit | 33.07 | 92.31 | 44.93 | 88.02 | 9.12 | 97.77 | 33.06 | 92.17 | 30.05 | 92.57 | 87.23 |
| Mahalanobis | 14.03 | 96.69 | 43.22 | 86.68 | 18.13 | 93.30 | 11.97 | 97.10 | 21.84 | 93.44 | 87.23 |
| ReAct | 27.59 | 93.54 | 44.01 | 88.05 | 7.53 | 97.61 | 31.01 | 92.86 | 27.54 | 93.02 | 87.00 |
| ASH | 51.20 | 87.81 | 53.93 | 84.22 | 19.95 | 95.92 | 42.99 | 90.23 | 42.02 | 89.55 | 86.20 |
| GEN | 41.51 | 90.34 | 46.18 | 87.91 | 8.21 | 98.26 | 38.31 | 91.28 | 33.55 | 91.95 | 87.23 |
| KNN | 22.69 | 95.01 | 39.34 | 89.28 | 9.92 | 97.92 | 20.75 | 96.02 | 23.18 | 94.56 | 87.23 |
| VIM | 13.68 | 97.01 | 33.87 | 91.45 | 5.93 | 98.15 | 13.45 | 97.12 | 16.73 | 95.93 | 87.23 |
| LogitNorm | 46.07 | 87.41 | 49.03 | 85.96 | 15.96 | 96.30 | 47.09 | 87.64 | 39.54 | 89.33 | 86.09 |
| | | | | | With A2D Training and NP-Mix Outlier Synthesis | | | | | | |
| MSP++ | $29.42_{-9.69}$ | $90.73_{+1.95}$ | $40.02_{-6.62}$ | $88.08_{+1.68}$ | $13.34_{-3.99}$ | $96.43_{+0.44}$ | $28.16_{-11.75}$ | $91.63_{+2.53}$ | $27.74_{-8.01}$ | $91.72_{+1.65}$ | 86.89 |
| Energy++ | $24.52_{-8.43}$ | $93.96_{+1.48}$ | $36.49_{-8.44}$ | $89.67_{+1.72}$ | $6.96_{-1.14}$ | $97.53_{-0.17}$ | $22.92_{-10.14}$ | $94.41_{+2.13}$ | $22.72_{-7.01}$ | $93.89_{+1.29}$ | 86.89 |
| MaxLogit++ | $24.86_{-8.21}$ | $93.69_{+1.38}$ | $36.60_{-8.33}$ | $89.71_{+1.69}$ | $6.96_{-2.16}$ | $97.67_{-0.10}$ | $22.92_{-10.14}$ | $94.22_{+2.05}$ | $22.84_{-7.21}$ | $93.82_{+1.25}$ | 86.89 |
| Mahalanobis++ | $9.01_{-5.02}$ | $97.72_{+1.03}$ | $27.94_{-15.28}$ | $91.09_{+4.41}$ | $12.77_{-5.36}$ | $95.96_{+2.66}$ | $7.64_{-4.33}$ | $98.23_{+1.13}$ | $14.34_{-7.50}$ | $95.75_{+2.31}$ | 86.89 |
| ReAct++ | $21.09_{-6.50}$ | $94.72_{+1.18}$ | $37.51_{-6.50}$ | $89.66_{+1.61}$ | $7.30_{-0.23}$ | $97.38_{-0.23}$ | $20.64_{-10.37}$ | $95.01_{+2.15}$ | $21.63_{-5.91}$ | $94.19_{+1.17}$ | 86.66 |
| ASH++ | $27.82_{-23.38}$ | $93.17_{+5.36}$ | $38.43_{-15.50}$ | $89.52_{+5.30}$ | $6.84_{-13.11}$ | $98.23_{+2.31}$ | $23.03_{-19.96}$ | $94.45_{+4.22}$ | $24.03_{-17.99}$ | $93.84_{+4.29}$ | 86.20 |
| GEN++ | $25.66_{-15.85}$ | $93.50_{+3.16}$ | $37.40_{-8.78}$ | $91.19_{+3.28}$ | $5.25_{-2.96}$ | $98.98_{+0.72}$ | $24.63_{-13.68}$ | $94.28_{+3.00}$ | $23.24_{-10.31}$ | $94.49_{+2.54}$ | 86.89 |
| KNN++ | $15.05_{-7.64}$ | $96.96_{+1.95}$ | $33.06_{-6.28}$ | $91.92_{+2.64}$ | $5.47_{-4.45}$ | $98.97_{+1.05}$ | $13.45_{-7.30}$ | $97.25_{+1.23}$ | $16.76_{-6.42}$ | $96.28_{+1.72}$ | 86.89 |
| VIM++ | $9.24_{-4.44}$ | $98.04_{+1.03}$ | $26.45_{-7.42}$ | $92.34_{+0.89}$ | $5.36_{-0.57}$ | $98.09_{-0.06}$ | $6.04_{-7.41}$ | $98.56_{+1.44}$ | $11.77_{-4.96}$ | $96.76_{+0.83}$ | 86.89 |
| LogitNorm++ | $29.53_{-16.54}$ | $91.44_{+4.03}$ | $30.22_{-18.81}$ | $91.37_{+5.41}$ | $13.23_{-2.73}$ | $96.81_{+0.51}$ | $25.88_{-21.21}$ | $93.16_{+5.52}$ | $24.72_{-14.82}$ | $93.20_{+3.87}$ | 86.89 |

Table 8: **Multimodal Far-OOD Detection** using video and optical flow, with **Kinetics-600** as ID.

| Methods | HMDB51 | | UCF101 | | EPIC-Kitchens | | HAC | | Average | | ID ACC ↑ |
|---|---|---|---|---|---|---|---|---|---|---|---|
| | FPR95↓ | AUROC↑ | FPR95↓ | AUROC↑ | FPR95↓ | AUROC↑ | FPR95↓ | AUROC↑ | FPR95↓ | AUROC↑ | |
| | | | | | Without A2D Training | | | | | | |
| MSP | 66.83 | 75.64 | 67.32 | 71.13 | 43.37 | 86.66 | 56.17 | 79.50 | 58.42 | 78.23 | 73.14 |
| Energy | 72.64 | 71.75 | 70.12 | 71.49 | 43.66 | 82.05 | 61.50 | 74.99 | 61.98 | 75.07 | 73.14 |
| MaxLogit | 72.06 | 73.68 | 70.47 | 71.96 | 39.84 | 84.76 | 57.95 | 77.27 | 60.08 | 76.92 | 73.14 |
| Mahalanobis | 89.01 | 56.24 | 81.85 | 64.49 | 93.12 | 41.59 | 95.88 | 48.85 | 89.97 | 52.79 | 73.14 |
| ReAct | 82.17 | 67.09 | 78.44 | 67.64 | 53.49 | 78.07 | 75.11 | 70.04 | 72.30 | 70.71 | 73.27 |
| ASH | 71.62 | 76.66 | 69.36 | 72.38 | 34.38 | 88.05 | 47.85 | 83.49 | 55.80 | 80.15 | 72.20 |
| GEN | 68.47 | 78.43 | 64.80 | 73.97 | 36.81 | 85.11 | 49.53 | 83.67 | 54.90 | 80.30 | 73.14 |
| KNN | 71.08 | 78.84 | 68.62 | 74.33 | 41.83 | 82.32 | 57.00 | 82.53 | 59.63 | 79.51 | 73.14 |
| VIM | 72.25 | 71.88 | 70.72 | 70.58 | 43.14 | 82.69 | 59.48 | 75.46 | 61.40 | 75.15 | 73.14 |
| LogitNorm | 66.48 | 79.00 | 63.79 | 75.10 | 39.03 | 85.27 | 54.22 | 81.83 | 55.88 | 80.30 | 74.47 |
| | | | | | With A2D Training and NP-Mix Outlier Synthesis | | | | | | |
| MSP++ | $62.76_{-4.07}$ | $77.53_{+1.89}$ | $66.33_{-0.99}$ | $73.13_{+2.00}$ | $52.84_{+9.47}$ | $83.78_{-2.88}$ | $53.37_{-2.80}$ | $80.82_{+1.32}$ | $58.83_{+0.41}$ | $78.82_{+0.59}$ | 73.67 |
| Energy++ | $63.27_{-9.37}$ | $74.17_{+2.42}$ | $67.20_{-2.92}$ | $74.50_{+3.01}$ | $34.07_{-9.59}$ | $87.49_{+5.44}$ | $56.69_{-4.81}$ | $80.20_{+5.21}$ | $55.31_{-6.67}$ | $79.09_{+4.02}$ | 73.67 |
| MaxLogit++ | $61.70_{-10.36}$ | $75.95_{+2.27}$ | $67.46_{-3.01}$ | $74.98_{+3.02}$ | $32.24_{-7.60}$ | $88.74_{+3.98}$ | $52.52_{-5.43}$ | $81.95_{+4.68}$ | $53.48_{-6.60}$ | $80.41_{+3.49}$ | 73.67 |
| Mahalanobis++ | $85.93_{-3.08}$ | $57.32_{+1.08}$ | $81.96_{+0.11}$ | $62.14_{-2.35}$ | $89.08_{-4.04}$ | $50.37_{+8.78}$ | $95.38_{-0.50}$ | $48.55_{-0.30}$ | $88.09_{-1.87}$ | $54.60_{+1.81}$ | 73.67 |
| ReAct++ | $73.00_{-9.17}$ | $70.09_{+3.00}$ | $76.09_{-2.35}$ | $71.40_{+3.76}$ | $39.06_{-14.43}$ | $85.42_{+7.35}$ | $70.09_{-5.02}$ | $76.04_{+6.00}$ | $64.56_{-7.74}$ | $75.74_{+5.03}$ | 73.65 |
| ASH++ | $63.65_{-7.97}$ | $79.14_{+2.48}$ | $62.15_{-7.21}$ | $75.32_{+2.94}$ | $36.09_{+1.71}$ | $88.69_{+0.64}$ | $46.47_{-1.38}$ | $85.21_{+1.72}$ | $52.09_{-3.71}$ | $82.09_{+1.94}$ | 72.53 |
| GEN++ | $61.67_{-6.80}$ | $80.63_{+2.20}$ | $57.03_{-7.77}$ | $77.97_{+4.00}$ | $40.09_{+3.28}$ | $85.10_{-0.01}$ | $41.97_{-7.56}$ | $86.98_{+3.31}$ | $50.19_{-4.71}$ | $82.67_{+2.37}$ | 73.67 |
| KNN++ | $64.60_{-6.48}$ | $80.28_{+1.44}$ | $67.44_{-1.18}$ | $77.95_{+3.62}$ | $37.65_{-4.18}$ | $83.55_{+1.23}$ | $53.63_{-3.37}$ | $83.34_{+0.81}$ | $55.83_{-3.80}$ | $81.28_{+1.77}$ | 73.67 |
| VIM++ | $63.25_{-9.00}$ | $73.96_{+2.08}$ | $66.31_{-4.41}$ | $74.08_{+3.50}$ | $34.45_{-8.69}$ | $87.67_{+4.98}$ | $53.82_{-5.66}$ | $81.06_{+5.60}$ | $54.46_{-6.94}$ | $79.19_{+4.04}$ | 73.67 |
| LogitNorm++ | $67.67_{+1.19}$ | $79.15_{+0.15}$ | $59.14_{-4.65}$ | $77.91_{+2.81}$ | $31.57_{-7.46}$ | $88.76_{+3.49}$ | $57.51_{+3.29}$ | $78.92_{-2.91}$ | $53.97_{-1.91}$ | $81.19_{+0.89}$ | 70.81 |

**Multimodal Near-OOD Detection with Other Combination of Modalities.** We demonstrate the effectiveness of *A2D* and *NP-Mix* across various combinations of modalities, not limited to video and optical flow, as shown in Tab. 9. The performance of different baseline algorithms improves significantly with *A2D* training and *NP-Mix* outlier synthesis, regardless of whether the input modalities are video-audio, flow-audio, or video-audio-flow combinations.

# F More Ablations

**Compared with Other Multimodal Tasks.** We compare *A2D* and *NP-Mix* with other multimodal self-supervised training tasks, including Contrastive Loss [35], Relative Norm Alignment (RNA) Loss [51], Cross-modal Distillation [27], and Cross-modal Translation [20], as shown in Tab. 10. While contrastive loss demonstrates effectiveness in enhancing OOD performance, other tasks significantly decrease the performance. *A2D* and *NP-Mix* show substantial superiority over other multimodal self-supervised tasks.

**Influences of $N$ and $\alpha$ in *NP-Mix*.** In this section, we investigate the parameter sensitivity of *NP-Mix* on the HMDB51 25/26 dataset. For the Nearest Neighbor parameter $N$, we test values of 1, 2, 3, and 4. Regarding the Mixup parameter $\alpha$, we evaluate values of 2.0, 4.0, and 10.0. As shown

Table 9: **Multimodal Near-OOD Detection** using different combination of modalities.

| Methods | Modality | | | EPIC-Kitchens 4/4 | | | Kinetics-600 129/100 | | |
|---|---|---|---|---|---|---|---|---|---|
| | Video | Audio | Flow | FPR95↓ | AUROC↑ | ID ACC ↑ | FPR95↓ | AUROC↑ | ID ACC ↑ |
| | | | | **Without A2D Training** | | | | | |
| Energy | ✓ | ✓ | | 69.59 | 73.58 | 71.08 | 65.69 | 76.68 | 81.44 |
| ASH | ✓ | ✓ | | 75.75 | 63.89 | 62.69 | 63.32 | 78.04 | 80.86 |
| GEN | ✓ | ✓ | | 69.22 | 69.04 | 71.08 | 66.18 | 77.65 | 81.44 |
| KNN | ✓ | ✓ | | 97.20 | 31.97 | 71.08 | 77.38 | 64.36 | 81.44 |
| VIM | ✓ | ✓ | | 92.35 | 37.10 | 71.08 | 65.77 | 76.69 | 81.44 |
| | | | | **With A2D Training and NP-Mix Outlier Synthesis** | | | | | |
| Energy++ | ✓ | ✓ | | $67.72_{-1.87}$ | $72.66_{-0.92}$ | 69.40 | $64.12_{-1.57}$ | $77.15_{+0.47}$ | 81.23 |
| ASH++ | ✓ | ✓ | | $75.93_{+0.18}$ | $67.83_{+3.94}$ | 60.44 | $62.67_{-0.65}$ | $78.03_{-0.01}$ | 79.80 |
| GEN++ | ✓ | ✓ | | $67.35_{-1.87}$ | $72.66_{+3.62}$ | 69.40 | $62.20_{-3.98}$ | $78.01_{+0.36}$ | 81.23 |
| KNN++ | ✓ | ✓ | | $79.66_{-17.54}$ | $65.52_{+33.55}$ | 69.40 | $73.87_{-3.51}$ | $66.09_{+1.73}$ | 81.23 |
| VIM++ | ✓ | ✓ | | $77.80_{-14.55}$ | $57.50_{+20.40}$ | 69.40 | $63.65_{-2.12}$ | $77.23_{+0.54}$ | 81.23 |
| | | | | **Without A2D Training** | | | | | |
| Energy | | ✓ | ✓ | 73.88 | 62.59 | 67.91 | 77.76 | 65.07 | 56.18 |
| ASH | | ✓ | ✓ | 78.36 | 60.74 | 62.13 | 77.27 | 65.43 | 54.47 |
| GEN | | ✓ | ✓ | 74.25 | 61.48 | 67.91 | 77.46 | 64.73 | 56.18 |
| KNN | | ✓ | ✓ | 89.55 | 45.84 | 67.91 | 92.02 | 50.70 | 56.18 |
| VIM | | ✓ | ✓ | 84.33 | 45.99 | 67.91 | 77.78 | 65.09 | 56.18 |
| | | | | **With A2D Training and NP-Mix Outlier Synthesis** | | | | | |
| Energy++ | | ✓ | ✓ | $70.90_{-2.98}$ | $68.64_{+6.05}$ | 68.47 | $76.11_{-1.65}$ | $65.55_{+0.48}$ | 55.90 |
| ASH++ | | ✓ | ✓ | $72.57_{-5.79}$ | $66.20_{+5.46}$ | 60.45 | $75.68_{-1.59}$ | $65.86_{+0.43}$ | 54.55 |
| GEN++ | | ✓ | ✓ | $73.13_{-1.12}$ | $66.38_{+4.90}$ | 68.47 | $74.48_{-2.98}$ | $65.33_{+0.60}$ | 55.90 |
| KNN++ | | ✓ | ✓ | $82.09_{-7.46}$ | $58.14_{+12.30}$ | 68.47 | $89.98_{-2.04}$ | $53.94_{+3.24}$ | 55.90 |
| VIM++ | | ✓ | ✓ | $83.40_{-0.93}$ | $49.40_{+3.41}$ | 68.47 | $76.13_{-1.65}$ | $65.57_{+0.48}$ | 55.90 |
| | | | | **Without A2D Training** | | | | | |
| Energy | ✓ | ✓ | ✓ | 69.22 | 72.39 | 73.13 | 66.42 | 76.60 | 80.33 |
| ASH | ✓ | ✓ | ✓ | 70.52 | 69.70 | 68.10 | 63.48 | 78.11 | 79.54 |
| GEN | ✓ | ✓ | ✓ | 70.34 | 70.99 | 73.13 | 64.24 | 77.54 | 80.33 |
| KNN | ✓ | ✓ | ✓ | 80.22 | 60.56 | 73.13 | 75.03 | 65.97 | 80.33 |
| VIM | ✓ | ✓ | ✓ | 76.12 | 59.03 | 73.13 | 66.38 | 76.59 | 80.33 |
| | | | | **With A2D Training and NP-Mix Outlier Synthesis** | | | | | |
| Energy++ | ✓ | ✓ | ✓ | $62.69_{-6.53}$ | $74.95_{+2.56}$ | 71.46 | $63.81_{-2.61}$ | $77.89_{+1.29}$ | 80.82 |
| ASH++ | ✓ | ✓ | ✓ | $69.78_{-0.74}$ | $69.49_{-0.21}$ | 67.72 | $61.22_{-2.26}$ | $78.57_{+0.46}$ | 80.05 |
| GEN++ | ✓ | ✓ | ✓ | $63.62_{-6.72}$ | $73.94_{+2.95}$ | 71.46 | $63.55_{-0.69}$ | $77.92_{+0.38}$ | 80.82 |
| KNN++ | ✓ | ✓ | ✓ | $68.47_{-11.75}$ | $67.87_{+7.31}$ | 71.46 | $71.46_{-3.57}$ | $68.87_{+2.90}$ | 80.82 |
| VIM++ | ✓ | ✓ | ✓ | $73.51_{-2.61}$ | $59.57_{+0.54}$ | 71.46 | $63.42_{-2.96}$ | $77.90_{+1.31}$ | 80.82 |

Table 10: Ablation on **Multimodal Training Tasks** for Near-OOD Detection on HMDB51 dataset.

| Methods | Baseline | | Contrastive Loss | | RNA Loss | | Cross-modal Distillation | | Cross-modal Translation | | A2D+NP-Mix (ours) | |
|---|---|---|---|---|---|---|---|---|---|---|---|---|
| | FPR95↓ | AUROC↑ | FPR95↓ | AUROC↑ | FPR95↓ | AUROC↑ | FPR95↓ | AUROC↑ | FPR95↓ | AUROC↑ | FPR95↓ | AUROC↑ |
| Energy | 43.36 | 87.46 | 39.65 | 88.72 | 55.55 | 84.33 | 53.81 | 85.18 | 44.44 | 87.98 | 36.38 | 88.91 |
| GEN | 43.79 | 87.49 | 41.61 | 89.24 | 58.61 | 83.18 | 55.34 | 85.10 | 48.58 | 87.66 | 35.95 | 89.78 |
| KNN | 42.92 | 88.46 | 37.69 | 89.27 | 47.49 | 84.72 | 43.79 | 86.84 | 39.43 | 88.75 | 33.77 | 90.05 |
| VIM | 36.82 | 88.06 | 36.38 | 88.01 | 43.14 | 87.16 | 39.87 | 87.83 | 39.43 | 88.32 | 34.64 | 88.80 |

in Fig. 10 and Fig. 11, *NP-Mix* demonstrates robustness across different parameter settings and yields substantial enhancements in OOD performance for all cases.

**Different Architectures.** In this section, we demonstrate that the strong performance of training with *A2D* and *NP-Mix* is maintained across different network architectures. We replaced the architecture of the video backbone with Inflated 3D ConvNet (I3D) [7] and the optical flow backbone with Temporal Segment Network (TSN) [64]. As shown in Tab. 12, *A2D* and *NP-Mix* consistently deliver significant improvement in OOD detection performance with these new architectures. Furthermore, we replaced the architecture of the video backbone architecture with Video Swin Transformer (Swin-T) [45] and the audio backbone with Audio Spectrogram Transformer (AST) [25]. As illustrated in Tab. 13, *A2D* and *NP-Mix* also consistently achieve significant improvement in OOD detection performance using Transformer-based architectures.

| | FPR95↓ | AUROC↑ |
|---|---|---|
| Video-only (MSP) | 60.78 | 84.39 |
| Flow-only (MSP) | 70.37 | 72.97 |
| Video-only (Energy) | 64.05 | 83.14 |
| Flow-only (Energy) | 71.46 | 75.51 |
| Video-only (Mahalanobis) | 51.20 | 81.25 |
| Flow-only (Mahalanobis) | 89.98 | 59.38 |
| Video-only (MSP+Energy) | 63.18 | 84.19 |
| Video-only (MSP+Mahalanobis) | 41.61 | 86.44 |
| Video-only (Energy+Mahalanobis) | 44.44 | 86.62 |
| Flow-only (MSP+Energy) | 70.15 | 73.40 |
| Flow-only (MSP+Mahalanobis) | 66.88 | 73.59 |
| Flow-only (Energy+Mahalanobis) | 65.36 | 74.69 |
| A2D+NP-Mix (ours, Energy) | **36.38** | **88.91** |

Table 11: Ablation on the ensemble of different OOD scores on a single modality.

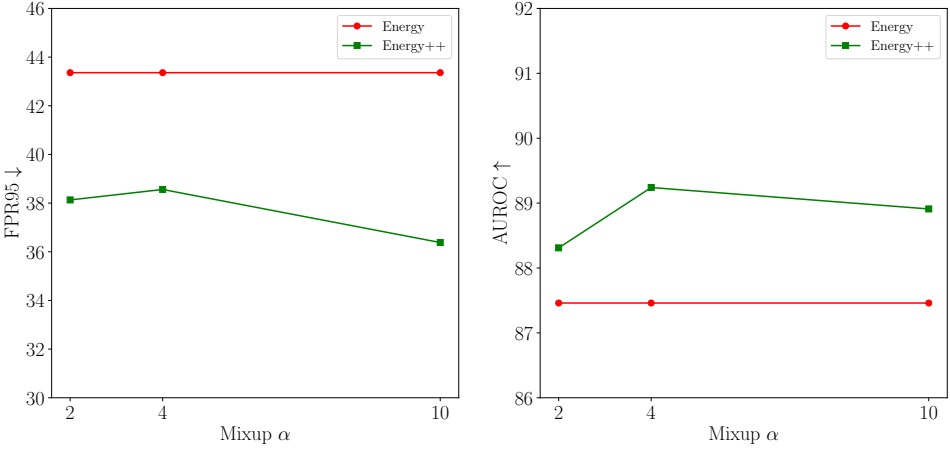

Figure 10: Influences of Mixup $\alpha$ for OOD performance in *NP-Mix*.

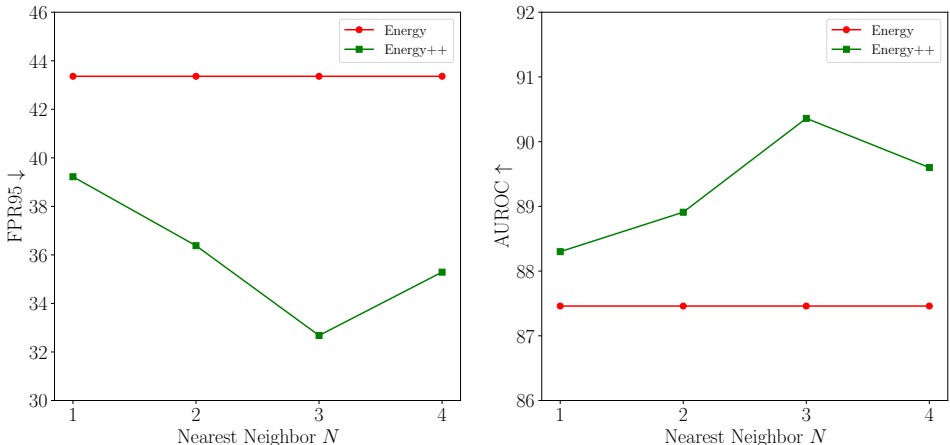

Figure 11: Influences of Nearest Neighbor $N$ for OOD performance in *NP-Mix*.

**Ensemble of Multiple Unimodal OOD Methods.** In this section, we add evaluations on HMDB51 25/26 for the ensemble of multiple unimodal OOD methods for each modality to demonstrate the importance of studying the multimodal OOD detection problem. We first evaluate the ensemble of different OOD scores on a single modality. We choose three scores for the ensemble: probability space (MSP), logit space (Energy), and feature space (Mahalanobis). For all scores, we normalize their values between 0 and 1 and calculate the

|  | FPR95↓ | AUROC↑ |
|---|---|---|
| Video (MSP) + Flow (MSP) | 50.98 | 85.40 |
| Video (Energy) + Flow (Energy) | 49.89 | 85.38 |
| Video (Mahalanobis) + Flow (Mahalanobis) | 52.07 | 81.27 |
| Video (MSP) + Flow (Energy) | 46.62 | 86.25 |
| Video (Energy) + Flow (MSP) | 50.98 | 83.69 |
| Video (MSP) + Flow (Mahalanobis) | 57.30 | 84.68 |
| Video (Mahalanobis) + Flow (MSP) | 49.02 | 82.92 |
| Video (Mahalanobis) + Flow (Energy) | 47.71 | 83.51 |
| Video (Energy) + Flow (Mahalanobis) | 59.91 | 81.96 |
| A2D+NP-Mix (ours, Energy) | **36.38** | **88.91** |

Table 14: Ablation on the ensemble of different OOD scores on different modalities.

ensemble score as: score = $\alpha$ * $score_1$ + $(1 - \alpha)$ * $score_2$. For $\alpha$, we do a grid search from 0.1 to 0.9 with a 0.1 interval and report the one with the best performance. As shown in Tab. 11, combining MSP or Energy with Mahalanobis can bring significant improvement, especially for video. However, there is still a large gap compared with our proposed solution, demonstrating the importance of studying the multimodal OOD detection problem. We then evaluate the ensemble of OOD scores on different modalities and calculate the ensemble score as: score = $\alpha$ * $score_{video}$ + $(1 - \alpha)$ * $score_{flow}$. For $\alpha$, we also do a grid search from 0.1 to 0.9 with a 0.1 interval and report the one with the best performance. As shown in Tab. 14, combining more modalities always brings performance improvements, but still has a large gap compared with our proposed solution, further demonstrating the importance of studying multimodal OOD detection problems.

Table 12: **Multimodal Near-OOD Detection** using video and flow on HMDB51 25/26 with I3D and TSN backbones.

| Methods | HMDB51 25/26 | | |
|---|---|---|---|
| | FPR95↓ | AUROC↑ | ID ACC ↑ |
| **Without A2D Training** | | | |
| MSP | 51.42 | 85.00 | 88.02 |
| Energy | 50.76 | 85.93 | 88.02 |
| MaxLogit | 50.98 | 85.95 | 88.02 |
| Mahalanobis | 60.13 | 79.00 | 88.02 |
| ReAct | 44.88 | 86.64 | 87.80 |
| ASH | 59.26 | 85.95 | 88.02 |
| GEN | 53.38 | 86.13 | 88.02 |
| KNN | 43.57 | 88.66 | 88.02 |
| VIM | 41.61 | 86.58 | 88.02 |
| LogitNorm | 46.62 | 86.03 | 88.45 |
| **With A2D Training and NP-Mix** | | | |
| MSP++ | $40.09_{-11.33}$ | $87.51_{+2.51}$ | 90.63 |
| Energy++ | $36.60_{-14.16}$ | $87.48_{+1.55}$ | 90.63 |
| MaxLogit++ | $36.60_{-14.38}$ | $87.69_{+1.74}$ | 90.63 |
| Mahalanobis++ | $54.68_{-5.45}$ | $81.36_{+2.36}$ | 90.63 |
| ReAct++ | $39.22_{-5.66}$ | $87.11_{+0.47}$ | 90.85 |
| ASH++ | $45.32_{-13.94}$ | $87.02_{+1.07}$ | 87.80 |
| GEN++ | $36.60_{-16.78}$ | $88.49_{+2.36}$ | 90.63 |
| KNN++ | $37.25_{-6.32}$ | $88.51_{-0.15}$ | 90.63 |
| VIM++ | $39.00_{-2.61}$ | $85.70_{-0.88}$ | 90.63 |
| LogitNorm++ | $39.00_{-7.62}$ | $87.79_{+1.76}$ | 87.58 |

Table 13: **Multimodal Near-OOD Detection** using video and audio on EPIC-Kitchens 4/4 with Swin-T and AST backbones.

| Methods | EPIC-Kitchens 4/4 | | |
|---|---|---|---|
| | FPR95↓ | AUROC↑ | ID ACC ↑ |
| **Without A2D Training** | | | |
| MSP | 87.87 | 53.67 | 61.01 |
| Energy | 83.40 | 56.44 | 61.01 |
| MaxLogit | 83.40 | 56.31 | 61.01 |
| Mahalanobis | 92.72 | 54.12 | 61.01 |
| ReAct | 83.21 | 56.45 | 60.45 |
| ASH | 94.96 | 54.23 | 52.61 |
| GEN | 86.38 | 54.37 | 61.01 |
| KNN | 91.04 | 52.03 | 61.01 |
| VIM | 85.45 | 56.68 | 61.01 |
| LogitNorm | 81.34 | 59.37 | 58.40 |
| **With A2D Training and NP-Mix** | | | |
| MSP++ | $77.24_{-10.63}$ | $67.02_{+13.35}$ | 62.31 |
| Energy++ | $69.78_{-13.62}$ | $68.89_{+12.45}$ | 62.31 |
| MaxLogit++ | $69.96_{-13.44}$ | $68.99_{+12.68}$ | 62.31 |
| Mahalanobis++ | $89.93_{-2.79}$ | $56.58_{+2.46}$ | 62.31 |
| ReAct++ | $70.52_{-12.69}$ | $68.72_{+12.27}$ | 62.13 |
| ASH++ | $89.74_{-5.22}$ | $56.65_{+2.42}$ | 14.93 |
| GEN++ | $73.13_{-13.25}$ | $69.46_{+15.09}$ | 62.31 |
| KNN++ | $77.05_{-13.99}$ | $63.35_{+11.32}$ | 62.31 |
| VIM++ | $71.83_{-13.62}$ | $67.87_{+11.19}$ | 62.31 |
| LogitNorm++ | $82.84_{+1.50}$ | $57.54_{-1.83}$ | 58.58 |

**Robustness under Missing-modalities.** In our framework, we train a classifier for each modality to get predictions from each modality separately. By default, we use the predictions obtained from the combined embeddings of all modalities to calculate the OOD score. However, when one modality is missing, we can use the predictions from the remaining modality to calculate the OOD score. We add evaluations on HMDB51 25/26 under this challenging condition and use Energy as the OOD score. As shown in Tab. 15, when one modality is missing, the performance drops a little, especially in the case when the video is missing (A2D+NP-Mix (Flow)). However, compared with training on one modality alone (Video-only and Flow-only), training with A2D and NP-Mix can also bring significant improvements for each modality when another modality is missing. For example, A2D+NP-Mix (Video), the case when optical flow is missing, yields a 16.56% relative improvement on FPR95 compared with Video-only. This underscores the importance of cross-modal training for multimodal OOD detection.

| | FPR95↓ | AUROC↑ |
|---|---|---|
| Video-only | 64.05 | 83.14 |
| Flow-only | 71.46 | 75.51 |
| A2D+NP-Mix (Video) | 47.49 | 86.48 |
| A2D+NP-Mix (Flow) | 66.01 | 77.23 |
| A2D+NP-Mix | 36.38 | 88.91 |

Table 15: Ablation on the robustness under missing-modalities.

**Beyond Action Recognition Task.** To further demonstrate the versatility of the proposed A2D training, we add another task on 3D semantic segmentation using LiDAR point cloud and RGB images. We evaluate on SemanticKITTI [3] dataset and set all vehicle classes as OOD classes. During training, we set the labels of OOD classes to void and ignore them. During inference, we aim to segment the known classes with high Intersection over Union (IoU) score, and detect OOD classes as unknown. We adopt three metrics for evaluation, including FPR95, AUROC, and $mIOU_c$ (mean Intersection over Union for known classes). We use ResNet-34 [29] and SalsaNext [12] as the backbones of the camera stream and LiDAR stream. We compare our A2D with basic LiDAR-only and Late Fusion, as well as two multimodal 3D Semantic Segmentation baselines PMF [72] and XMUDA [33]. As shown in Tab. 16, our A2D also demonstrates strong performance under this new task (3D Semantic Segmentation) with different combinations of modalities (LiDAR point cloud and RGB images).

| | FPR95↓ | AUROC↑ | $mIOU_c$ ↑ |
|---|---|---|---|
| LiDAR-only | 57.78 | 84.76 | 59.81 |
| Late Fusion | 53.43 | 86.98 | 61.43 |
| PMF | 51.57 | 88.13 | 61.38 |
| XMUDA | 55.49 | 89.99 | 61.45 |
| A2D | **49.02** | **91.12** | **61.98** |

Table 16: Ablation on the 3D semantic segmentation using LiDAR point cloud and RGB images.

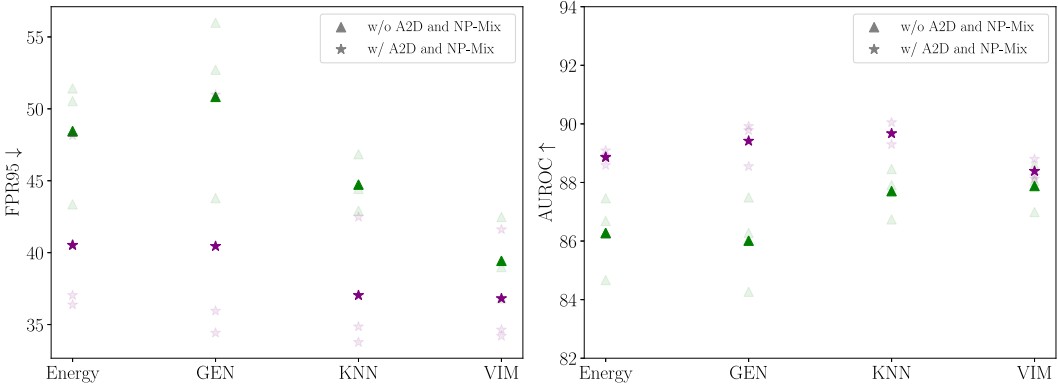

Figure 12: Experiments using three random seeds for Multimodal Near-OOD Detection on the HMDB51 25/26. Foreground points in bold show results averaged across three different seeds while background points, shown feint, indicate results from the underlying individual seeds.

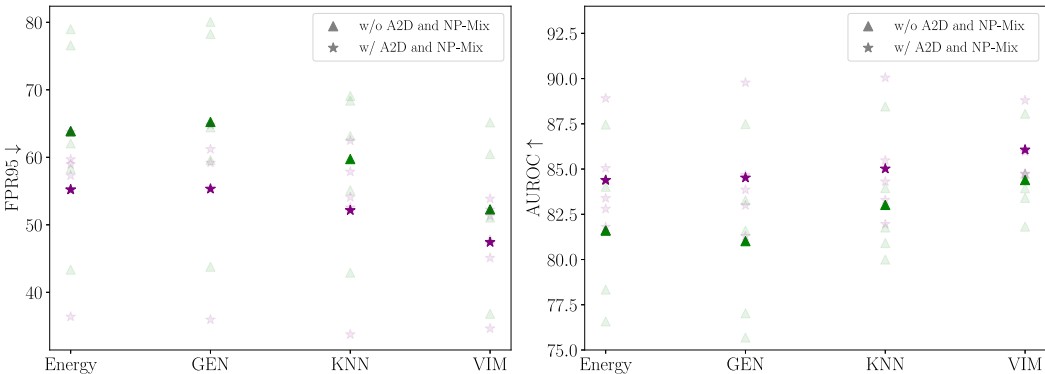

Figure 13: Experiments using five random dataset splits for Multimodal Near-OOD Detection on the HMDB51 25/26. Foreground points in bold show results averaged across five different splits while background points, shown feint, indicate results from the underlying individual splits.

# G   Statistical Significance Tests

**Different Random Seeds.** We run each experiment three times using different seeds for Multimodal Near-OOD Detection on the HMDB51 25/26 dataset in our MultiOOD benchmark, and then calculate the mean AUROC and FPR95 to demonstrate the statistical significance of our methods. As shown in Fig. 12, training with *A2D* and *NP-Mix* is statistically stable and significantly surpasses the baselines under different random seeds.

**Different Random Splits.** We run each experiment five times using different dataset splits for Multimodal Near-OOD Detection on the HMDB51 25/26 dataset in our MultiOOD benchmark, and then calculate the mean AUROC and FPR95 to demonstrate the statistical significance of our methods. As shown in Fig. 13, training with *A2D* and *NP-Mix* is statistically stable and surpasses the baselines significantly under different dataset splits.

