# OpenReview forum: "MultiOOD: Scaling Out-of-Distribution Detection for Multiple Modalities"
_NeurIPS.cc/2024/Conference — NeurIPS 2024 spotlight_

### Official Review · Reviewer_z2YR · 2024-06-15

**Soundness:** 2
**Presentation:** 3
**Contribution:** 2
**Rating:** 6
**Confidence:** 3

**Summary:**

This paper first proposes a MultiOOD benchmark for multimodal OOD detection. It then further proposes methods including A2D and NP-Mix for better tacking the multimodal OOD detection task.

**Strengths:**

1. I appreciate the formulation of the multimodal OOD detection benchmark.

2. This work presents extensive experiments and achieves good results.

**Weaknesses:**

(See the questions for more details.)

**Questions:**

Overall, I am now hesitating around the borderline for this paper and below are my concerns:

1. Generally, I believe that the relationship between this work and other multi-modality works can be very close. I thus suggest the authors to also discuss with other multi-modality works in their related works. I will also elaborate on this more in below concerns.

2. As for the benchmark, I believe that it is important for the evaluation to be conducted in a more statistically significant manner. In other words, from my perspective, the authors may consider to at least evaluate methods on different ID/OOD category splits and report standard deviation.

3. W.r.t. A2D in Sec 4.2, from my perspective, it seems that sometimes, the orders of non-ground-truth classes matter. For example, while for a cat input, both fox and truck are non-ground-truth classes, it seems natural for fox to have a higher softmax score than truck. Thus, I am curious, what if the discrepancy is maximized in an order-preserved manner. I appreciate if this can be performed as an ablation study to see whether this alternative of A2D can be more effective.

4. W.r.t. Sec 4.3, the authors claim that they get inspiration from [16]. I thus hope to see a more detailed comparison between the proposed method and the method in [16]. Besides, it seems to also worth comparing with Learning Placeholders for Open-Set Recognition CVPR 2021. If I am not wrong, it seems that the proposed NP-Mix is somehow similar to its data placeholder.

5. As for the evaluation, as mentioned in the first concern, this work is closedly related to those multi-modaity ones. Thus, to better validate the efficacy of A2D and NP-Mix, besides comparing with only uni-modal OOD methods, it is also suggeted to create baselines that combine uni-modal OOD methods and existing typical multi-modality methods outside the OOD area. This can lead the proposed method to be better evaluated.

**Limitations:**

This work has discussed its potential limitations.

---

> ### Author Rebuttal · Authors · 2024-08-04
>
> Thanks for your insightful reviews, and we appreciate your valuable suggestions! We address your concerns and questions as follows:
>
> >**Q1**: Discuss with other multi-modality works in related works.
>
> **A1**: Thanks for your insightful suggestion! As shown in **Table 10** in our main paper, we indeed compared A2D and NP-Mix with other multimodal self-supervised training tasks, including Contrastive Loss, Relative Norm Alignment, Cross-modal Distillation, and Cross-modal Translation. A2D and NP-Mix show substantial superiority over other multimodal tasks. We will add a subsection on Multimodal Learning in related work to discuss these works.
> ___
>
> >**Q2**: Evaluate methods on different ID/OOD category splits.
>
> **A2**: Thanks for your valuable comment! As shown in **Figure 13** in our main paper, we indeed evaluated our method on **five different ID/OOD category splits**. Training with our A2D and NP-Mix is statistically stable and surpasses the baselines significantly under different dataset splits. We also evaluated our method on **three different random seeds**, as shown in **Figure 12** in our main paper. Our method significantly surpasses the baselines under different random seeds.
> ___
>
> >**Q3**: W.r.t. A2D, the orders of non-ground-truth classes matter. what if the discrepancy is maximized in an order-preserved manner.
>
> **A3**: Thanks for your interesting idea! There are two small issues with the idea. One issue is the conflict object with A2D training. A2D aims to enlarge the prediction distance between different modalities for all non-ground-truth classes. If we want to preserve the order of predictions for different modalities, for example, we make the most similar non-ground-truth class have the second-largest prediction for both video and optical flow, the prediction distance between them will decrease as a result. The second issue is on the implementation. For each class, we need to manually select its most similar classes to define the order, which is complicated for datasets with a large number of classes.
>
> We implement this idea on EPIC-Kitchens 4/4 using video and optical flow and use Energy as OOD score. EPIC-Kitchens 4/4 has four ID classes:  ‘put’, ‘take’, ‘open’, ‘close’. We define 'put' and 'take' as paired classes (most similar) and ‘open’ and ‘close’ as another paired classes. Given an input sample, we preserve its prediction order for both modalities by making the paired classes have the second-largest prediction. As shown below, A2D with order-preserved prediction is better than w/o A2D, but with a large gap compared with the original A2D. Besides, because of the conflict object with A2D, the prediction discrepancy $l_{OOD}-l_{ID}$ for order-preserved A2D is significantly reduced compared with the original A2D.
>
> ||FPR95$\downarrow$|AUROC$\uparrow$|$l_{OOD}-l_{ID}$|
> |-|-|-|-|
> |w/o A2D|76.68|68.29|0.2696|
> |A2D (order-preserved)|73.13|69.39|0.2866|
> |A2D|**66.98**|**72.45**|0.3987|
> ___
>
> >**Q4**: Detailed comparison between the proposed method and the method in [16]. it seems to also worth comparing with Learning Placeholders for Open-Set Recognition CVPR 2021.
>
> **A4**: Thanks for your useful suggestion! The major difference between [16] and our NP-Mix is the availability of real outliers during training. In [16], few labeled outliers are available and they generate more synthesized outliers based on the labeled outliers and unlabeled data. However, in our case, we assume only ID data is available during training and we want to generate synthesized outliers using only ID data. Therefore, [16] can't be used in our case directly. We make a small modification to it and implement it as a baseline. Instead of randomly selecting one sample from real outliers and another sample from its nearest neighbor in unlabeled data for Mixup, we randomly select a sample from one ID class and another sample from its nearest neighbor in other ID classes for Mixup. We also include PROSER [a] as a baseline based on your suggestion. PROSER [a] conducts manifold mixup with the pairs from different classes without considering neighborhood information and will inject manifold intrusion problem, as shown in Figure 2 in the attached PDF in global response. Instead, our NP-Mix explores broader feature spaces by leveraging the information from nearest neighbor classes without noisy synthesized outliers injection and achieves the best performances.
>
> ||FPR95$\downarrow$|AUROC$\uparrow$|
> |-|-|-|
> |A2D+NNG-Mix [16]|38.78|88.07|
> |PROSER|44.01|87.52|
> |A2D+NP-Mix (ours)|**36.38**|**88.91**|
>
> [a] Zhou, et al. Learning placeholders for open-set recognition. In: CVPR, 2021
> ___
>
> >**Q5**: create baselines that combine uni-modal OOD methods and existing typical multi-modality methods outside the OOD area.
>
> **A5**: Thanks for your suggestions! We added evaluations on HMDB51 25/26 for the ensemble of multiple unimodal OOD methods for each modality to demonstrate the importance of studying MultiOOD. Due to space limits, we put the detailed results in the **global response** at the top of this page. The ensemble of multiple unimodal OOD methods always brings performance improvements, but still has a large gap compared with our multimodal solution (A2D+NP-Mix), further demonstrating the importance of studying MultiOOD.
>
> For baselines on typical multimodal methods outside the OOD area, we already compared A2D and NP-Mix with four multimodal self-supervised training tasks in Table 10 in our main paper. Here, we further include two multimodal baselines Gradient Blending [b] and SimMMDG [c]. A2D and NP-Mix show substantial superiority over other multimodal methods.
>
> ||FPR95$\downarrow$|AUROC$\uparrow$|
> |-|-|-|
> |Gradient Blending|42.92|87.28|
> |SimMMDG|42.05|87.91|
> |A2D+NP-Mix (ours)|**36.38**|**88.91**|
>
> [b] Wang, et al. What makes training multi-modal classification networks hard? In: CVPR, 2020
>
> [c] Dong, et al. SimMMDG: A simple and effective framework for multi-modal domain generalization. In: NeurIPS, 2023

---

> > ### Comment · Reviewer_z2YR · 2024-08-08
> >
> > Thank the authors for their response. My most concerns have been well-solved and I thus increase my rate from 4 to 6.

---

> > > ### Author Response · Authors · 2024-08-08
> > > **Thanks for recognizing our work and raising your rating to 6!**
> > >
> > > We are glad to hear that we have addressed most of your concerns and that you have raised your rating to 6! Thanks for spending a significant amount of time on our submission and giving lots of valuable and insightful suggestions, which make our paper even stronger! We will also include all added experiments and points in the final paper for better clarification.

---

### Official Review · Reviewer_gP94 · 2024-07-07

**Soundness:** 3
**Presentation:** 3
**Contribution:** 3
**Rating:** 6
**Confidence:** 4

**Summary:**

This paper introduces MultiOOD, a new benchmark for multimodal out-of-distribution (OOD) detection. The authors propose two new techniques: 1) Agree-to-Disagree (A2D), which encourages discrepancy between modality predictions during training, and 2) NP-Mix, a novel outlier synthesis method. Extensive experiments on the MultiOOD benchmark demonstrate significant improvements over existing unimodal OOD detection methods.

**Strengths:**

Originality:
-First benchmark for multimodal OOD detection (MultiOOD).
-A2D algorithm leveraging modality prediction discrepancy.
-Outlier synthesis method (NP-Mix).

Quality:
-Strong performance improvements over baselines.
-Thoughtful analysis of modality prediction discrepancy phenomenon.
-Comprehensive experiments on diverse datasets.

Clarity:
-Well-organized structure.
-Clear explanations of key concepts and methods.

**Weaknesses:**

Weaknesses:
1) Lacks important baselines (e.g. ensemble of multiple singleOOD methods for each modality). This also relates to the question of "why we should study MultiOOD?" The importance of studying MultiOOD and its practical impact on real-world applications could be further demonstrated.
2) Limited theoretical analysis of proposed methods. A2D and NP-Mix need more theoretical analysis.
3) The experiments are only conducted on the task of action recognition. In this case, the title "MultiOOD" seems over-claiming. Please specify the task in the title and the main text. In addition, it only focuses on specific modalities (video, optical flow, audio). This makes the scope of the paper somewhat limited.
5) Some figures (e.g. Fig. 4) could be improved for clarity, with consistent color-coding and same x-axis scales.

**Questions:**

1) Why "α > 1" can ensure the synthesized outliers reside in the intermediary space between two prototypes, rather than near the ID data? Please give more explanations.
2) How sensitive are the A2D and NP-Mix methods to hyperparameter choices?

**Limitations:**

The authors acknowledge some limitations, such as the focus on specific modalities (video, optical flow, audio) and action recognition tasks. However, they could further discuss potential limitations in scaling to a larger number of modalities or very different types of data.

---

> ### Author Rebuttal · Authors · 2024-08-04
>
> Thanks for your insightful reviews, and we appreciate your valuable suggestions! Please find the responses to your questions below:
>
> >**Q1**: Lacks important baselines (e.g. ensemble of multiple singleOOD methods for each modality). This also relates to the question of "why we should study MultiOOD?"
>
> **A1**: Thanks for your suggestions! We added evaluations on HMDB51 25/26 for the ensemble of multiple unimodal OOD methods for each modality to demonstrate the importance of studying MultiOOD. Due to space limits, we put the detailed results in the **global response** at the top of this page. The ensemble of multiple unimodal OOD methods always brings performance improvement, but still has a large gap compared with our multimodal solution (A2D+NP-Mix), further demonstrating the importance of studying MultiOOD.
> ___
> >**Q2**: Limited theoretical analysis of proposed methods. A2D and NP-Mix need more theoretical analysis.
>
> **A2**: Thanks for bringing up this point. Our A2D is based on the empirical observation of Modality Prediction Discrepancy on our MultiOOD benchmark. This discrepancy can be attributed to the unavailability of semantic information on OOD data during model training, stimulating each modality to generate conjectures based on its unique characteristics upon encountering OOD data during testing. We demonstrate that such discrepancy is highly correlated to the ultimate OOD performance. We also show through extensive experiments that A2D can amplify such discrepancy and enhance the efficacy of OOD detection. For NP-Mix, we give an additional analysis on decision boundaries, as shown in Figure 2 in attached PDF in global response. Vanilla Mixup causes manifold intrusion due to the injection of noisy synthesized outliers. Instead, our NP-Mix only generates synthesized outliers in the intermediary space between two classes, thus avoiding the manifold intrusion and helping the network learn better decision boundaries.
>
> In summary, Multimodal OOD Detection is in its very early era. Our MultiOOD benchmark aims to fill this gap and our proposed A2D+NP-Mix solution is mostly based on empirical observation from extensive experiments. More interesting solutions and theoretical analyses are expected in future work.
> ___
> >**Q3**: The experiments are only conducted on the task of action recognition. the title "MultiOOD" seems over-claiming. In addition, it only focuses on specific modalities (video, optical flow, audio). This makes the scope of the paper somewhat limited.
>
> **A3**: Thanks for your insightful comments! To further increase the scope of our paper and demonstrate the versatility of the proposed A2D training, we add another task of 3D Semantic Segmentation using LiDAR point cloud and RGB images. We evaluate on SemanticKITTI [a] dataset and set all vehicle classes as OOD classes. During training, we set the labels of OOD classes to void and ignore them. During inference, we aim to segment the known classes with high Intersection over Union (IoU) score, and detect OOD classes as unknown. We adopt three metrics for evaluation, including FPR95, AUROC, and $mIOU_c$ (mean Intersection over Union for known classes). We use ResNet-34 and SalsaNext [b] as the backbones of the camera stream and LiDAR stream. We compare our A2D with basic LiDAR-only and Late Fusion, as well as two multimodal 3D semantic segmentation baselines PMF [c] and XMUDA [d]. Our A2D also shows strong performance under this new task (**3D Semantic Segmentation**) with different combinations of modalities (**LiDAR point cloud and RGB images**). We will integrate these new benchmark results in our paper to further increase its scope.
>
> ||FPR95$\downarrow$|AUROC$\uparrow$|$mIOU_c\uparrow$|
> |-|-|-|-|
> |LiDAR-only|57.78|84.76|59.81|
> |Late Fusion|53.43|86.98|61.43|
> |PMF|51.57|88.13|61.38|
> |XMUDA|55.49|89.99|61.45|
> |A2D (ours)|**49.02** |**91.12**|**61.98**|
>
> [a] Behley, et al. Semantickitti: A dataset for semantic scene understanding of lidar sequences. In ICCV, 2021
>
> [b] Cortinhal, et al. Salsanext: Fast, uncertainty-aware semantic segmentation of lidar point clouds. In ISVC, 2020
>
> [c] Zhuang, et al. Perception-aware multi-sensor fusion for 3d lidar semantic segmentation. In ICCV, 2021
>
> [d] Jaritz, et al. xmuda: Cross-modal unsupervised domain adaptation for 3d semantic segmentation. In CVPR, 2020
> ___
> >**Q4**: Some figures (e.g. Fig. 4) could be improved for clarity, with consistent color-coding and same x-axis scales.
>
> **A4**: Thanks for your suggestion! We have improved the figures and will update them in the final version of the paper.
> ___
> >**Q5**: Why "$\alpha$ > 1" can ensure the synthesized outliers reside in the intermediary space between two prototypes, rather than near the ID data?
>
> **A5**: Given a $\lambda$ sampled from distribution Beta($\alpha$, $\alpha$), as shown in Figure 1 in the attached PDF in global response, when $\alpha$ < 1, $\lambda$ has a very high probability of being close to 0 or 1. As a result, the synthesized data Z = $\lambda$ * Z1 + ($1-\lambda$) * Z2 will be close to Z1 or Z2. Instead, when $\alpha$ > 1, $\lambda$ has the highest probability near 0.5. As a result, the synthesized data Z = $\lambda$ * Z1 + ($1-\lambda$) * Z2 will be in the intermediary space between Z1 and Z2.
> ___
>
> >**Q6**: How sensitive are the A2D and NP-Mix methods to hyperparameter choices?
>
> **A6**: For A2D, we analyzed the choices of different distance functions, as shown in Table 5 in our main paper. A2D training exhibits robustness across various distance functions. Regardless of the specific distance metric employed, substantial improvements are consistently observed compared to the baseline approach without A2D training.
>
> We investigated the parameter sensitivity of NP-Mix on Nearest Neighbor parameter N and Mixup parameter $\alpha$, as shown in Figure 10 and Figure 11 in our main paper. NP-Mix demonstrates robustness across different parameter settings and yields substantial enhancements in OOD performance for all cases.

---

> > ### Comment · Reviewer_gP94 · 2024-08-10
> > **Thank you for the rebuttal.**
> >
> > Thanks for the rebuttal. Most of my concerns are well-solved. The reviewer suggests adding these information to the revision to make the paper stronger. I will increase my rating to weak accept.

---

> > > ### Author Response · Authors · 2024-08-10
> > > **Thanks for recognizing our work and raising your rating to weak accept!**
> > >
> > > We are glad to hear that we have well-solved most of your concerns and that you have raised your rating to weak accept! Thanks for spending a significant amount of time on our submission and giving lots of valuable and insightful suggestions, which make our paper even stronger! We will also include all added experiments and points in the final paper for better clarification.

---

### Official Review · Reviewer_Lt2v · 2024-07-07

**Soundness:** 4
**Presentation:** 4
**Contribution:** 4
**Rating:** 8
**Confidence:** 4

**Summary:**

The paper introduces a novel OOD detection benchmark for multimodal data (called MultiOOD), covering diverse dataset sizes and modalities. Based on this benchmark, the authors first demonstrate the Modality Prediction Discrepancy phenomenon, which means that
the discrepancies of softmax predictions are shown to be negligible for in-distribution data (across different modalities) and significant for OOD data. Based on these observations, the authors introduce a novel Agree-to-Disagree (A2D) algorithm, which aims to enhance
such discrepancies during training. Additionally, the authors propose a new outlier synthesis algorithm NP-Mix that explores broader feature spaces and complements A2D to strengthen the OOD detection performance. Experimental validation is extensive and confirms the improvements in OOD detection due to the new proposed algorithms.

**Strengths:**

Paper's strengths:
- the paper has a significant scientific contribution, by introducing the first multi-modal benchmark for OOD detection
- besides this, it contributes to the improvement of OOD detection by introducing two new algorithms:  A2D algorithm and a new outlier synthesis algorithm NP-Mix that explores broader feature spaces and complements A2D
- the paper is clearly written and well-documented
- the review of the state of the art is comprehensive and covers most of the relevant work
- experimental validation is extensive and underline the improvements introduced by the A2D and NP-Mix algorithms for OOD detection

**Weaknesses:**

- Some more details are required regarding some aspects.

**Questions:**

Here are my concerns:
- In the video modality, what features do you extract to characterize the stream? Do you extract per-frame features, or per-video features.
- Same question for the audio modality
- You use 5 datasets, all of them having the video and optical flow modalities, but audio modality is missing from 2 of them. In this case, what protocol do you adopt? Do you ignore them and perform the audio analysis only on the remaining three?
- Is your approach robust when one modality is missing?

**Limitations:**

The identified limitations are: (i) the performance on Near-OOD benchmark with a large number of classes can be further improved; and (ii) the ID/OOD discrepancy could also be further improved. The paper does not represent any negative societal impact.

---

> ### Author Rebuttal · Authors · 2024-08-04
>
> Thanks for your insightful reviews and great support of our paper! We provide the responses to your questions as follows:
>
> >**Q1**: In the video modality, what features do you extract to characterize the stream? Do you extract per-frame features, or per-video features.
>
> **A1**: Thanks for your valuable question! We use the SlowFast [a] network as the backbone to extract per-video features. The SlowFast model involves a Slow pathway operating at a low frame rate to capture spatial semantics and a Fast pathway operating at a high frame rate to capture motion at fine temporal resolution.
>
> [a] Feichtenhofer, et al. Slowfast networks for video recognition. In: ICCV, 2019
> ___
>
> >**Q2**: Same question for the audio modality
>
> **A2**: We also extract per-video features for the audio modality. Each 10-second audio waveform is converted into one spectrogram and then inputted to the ResNet-18 audio encoder. Audios that are less than 10 seconds are padded to 10 seconds.
> ___
>
> >**Q3**: You use 5 datasets, all of them having the video and optical flow modalities, but audio modality is missing from 2 of them. In this case, what protocol do you adopt? Do you ignore them and perform the audio analysis only on the remaining three?
>
> **A3**: All datasets have video and optical flow modalities. Therefore, we create four Multimodal Near-OOD benchmarks and two Multimodal Far-OOD benchmarks using video and optical flow, as shown in Figure 2 in the paper. Only EPIC-Kitchens, HAC, and Kinetics have audio modality. We create two challenging Multimodal Near-OOD benchmarks using EPIC-Kitchens and Kinetics with different combination of modalities (video-audio, flow-audio, video-flow-audio), as shown in Table 9 in the paper.
> ___
>
> >**Q4**: Is your approach robust when one modality is missing?
>
> **A4**: Thanks for your interesting comment! In our framework, we also train a classifier for each modality to get predictions from each modality separately. By default, we use the predictions obtained from the combined embeddings of all modalities to calculate the OOD score. However, when one modality is missing, we can use the predictions from the remaining modality to calculate the OOD score. We added the evaluations on HMDB51 25/26 under this challenging condition as below. We use Energy as the OOD score.
>
> |   | FPR95$\downarrow$ | AUROC$\uparrow$ |
> |---------|----------|----------|
> | Video-only |      64.05    |     83.14    |
> | Flow-only |      71.46    |    75.51  |
> | A2D+NP-Mix (Video) |       47.49  |        86.48    |
> | A2D+NP-Mix (Flow) |     66.01     |     77.23  |
> | A2D+NP-Mix |    36.38     |        88.91   |
>
> When one modality is missing, the performance drops a little, especially when the video is missing (A2D+NP-Mix (Flow)). However, compared with training on one modality alone (Video-only and Flow-only), training with A2D and NP-Mix can also bring significant improvements for each modality when another modality is missing. For example, A2D+NP-Mix (Video), the case when optical flow is missing, yields a 16.56\% relative improvement on FPR95 compared with Video-only. This underscores the importance of multimodal training for Multimodal OOD Detection.

---

> > ### Comment · Reviewer_Lt2v · 2024-08-12
> > **Acknowledgement of Rebuttal**
> >
> > I want to thank the authors for addressing all my concerns.

---

> > > ### Author Response · Authors · 2024-08-12
> > >
> > > We are glad to hear that we have addressed all your concerns. Thanks again for your insightful reviews and great support of our paper! We will also include all added experiments and points in the final paper for better clarification.

---

### Author Rebuttal · Authors · 2024-08-04

We sincerely thank all the reviewers for their encouraging and insightful comments. We have carefully read through them and provided global and individual responses, respectively. In global responses here, we first add evaluations for a general question on the **ensemble of multiple unimodal OOD methods**. Then, we add our new findings of A2D on a new task (**3D Semantic Segmentation**) with different combinations of modalities (**LiDAR point cloud and RGB images**). We also attach a PDF with figures to better illustrate answers for Reviewer gP94's questions on "Why $\alpha$ > 1 can ensure the synthesized outliers reside in the intermediary space between two prototypes, rather than near the ID data?" and "more detailed analysis on NP-Mix".

>**Ensemble of multiple unimodal OOD methods**

Based on the suggestions from Reviewer gP94 and Reviewer z2YR, we added evaluations on HMDB51 25/26 for the ensemble of multiple unimodal OOD methods for each modality to demonstrate the importance of studying MultiOOD. We first evaluate the ensemble of different OOD scores on a single modality. Specifically, we choose three different OOD scores for ensemble: probability space (MSP), logit space (Energy), and feature space (Mahalanobis). For all three scores, we normalize their values between 0 and 1 and calculate the ensemble score as score = $\alpha$ * score\_1 + ($1-\alpha$) * score\_2. For $\alpha$, we do a grid search from 0.1 to 0.9 with a 0.1 interval and report the best performance. As shown below, combining MSP or Energy with Mahalanobis can bring significant improvement, especially for video. However, there is still a large gap compared with our best multimodal OOD detection solution (A2D+NP-Mix), demonstrating the importance of studying MultiOOD.

||FPR95$\downarrow$|AUROC$\uparrow$|
|-|-|-|
|Video-only (MSP)|60.78|84.39|
|Flow-only (MSP)|70.37|72.97|
|Video-only (Energy)|64.05|83.14|
|Flow-only (Energy)|71.46|75.51|
|Video-only (Mahalanobis)|51.20|81.25|
|Flow-only (Mahalanobis)|89.98|59.38|
|Video-only (MSP+Energy)|63.18|84.19|
|Video-only (MSP+Mahalanobis)|41.61|86.44|
|Video-only (Energy+Mahalanobis)|44.44|86.62|
|Flow-only (MSP+Energy)|70.15|73.40|
|Flow-only (MSP+Mahalanobis)|66.88|73.59|
|Flow-only (Energy+Mahalanobis)|65.36|74.69|
|A2D+NP-Mix (ours best)|**33.77**|**90.05**|

We then evaluate the ensemble of various OOD scores on different modalities and calculate the ensemble score as score = $\alpha$ * score\_video + ($1-\alpha$) * score\_flow. In this case, we use the Video-only and Flow-only models as above. For $\alpha$, we also do a grid search from 0.1 to 0.9 with a 0.1 interval and report the one with the best performance. As shown below, combining more modalities always brings performance improvements, but still has a large gap compared with our A2D+NP-Mix, further demonstrating the importance of studying MultiOOD.

||FPR95$\downarrow$|AUROC$\uparrow$|
|-|-|-|
|Video (MSP) + Flow (MSP)|50.98|85.40|
|Video (Energy) + Flow (Energy)|49.89|85.38|
|Video (Mahalanobis) + Flow (Mahalanobis)|52.07|81.27|
|Video (MSP) + Flow (Energy)|46.62|86.25|
|Video (Energy) + Flow (MSP)|50.98|83.69|
|Video (MSP) + Flow (Mahalanobis)|57.30|84.68|
|Video (Mahalanobis) + Flow (MSP)|49.02|82.92|
|Video (Mahalanobis) + Flow (Energy)|47.71|83.51|
|Video (Energy) + Flow (Mahalanobis)|59.91|81.96|
|A2D+NP-Mix (ours best)|**33.77**|**90.05**|
___

>**Scope of the paper beyond action recognition using video, optical flow, and audio**

Based on the suggestions from Reviewer gP94, we add another task of 3D Semantic Segmentation using LiDAR point cloud and RGB images, to further increase the scope of our paper and demonstrate the versatility of the proposed A2D training. We evaluate on SemanticKITTI [a] dataset and set all vehicle classes as OOD classes. During training, we set the labels of OOD classes to void and ignore them. During inference, we aim to segment the known classes with high Intersection over Union (IoU) score, and detect OOD classes as unknown. We adopt three metrics for evaluation, including FPR95, AUROC, and $mIOU_c$ (mean Intersection over Union for known classes). We use ResNet-34 and SalsaNext [b] as the backbones of the camera stream and LiDAR stream. We compare our A2D with basic LiDAR-only and Late Fusion, as well as two multimodal 3D semantic segmentation baselines PMF [c] and XMUDA [d]. Our A2D also shows strong performance under this new task (**3D Semantic Segmentation**) with different combinations of modalities (**LiDAR point cloud and RGB images**). We will integrate these new benchmark results in our paper to further increase its scope.

||FPR95$\downarrow$|AUROC$\uparrow$|$mIOU_c\uparrow$|
|-|-|-|-|
|LiDAR-only|57.78|84.76|59.81|
|Late Fusion|53.43|86.98|61.43|
|PMF|51.57|88.13|61.38|
|XMUDA|55.49|89.99|61.45|
|A2D (ours)|**49.02** |**91.12**|**61.98**|

[a] Behley, et al. Semantickitti: A dataset for semantic scene understanding of lidar sequences. In ICCV, 2021

[b] Cortinhal, et al. Salsanext: Fast, uncertainty-aware semantic segmentation of lidar point clouds. In ISVC, 2020

[c] Zhuang, et al. Perception-aware multi-sensor fusion for 3d lidar semantic segmentation. In ICCV, 2021

[d] Jaritz, et al. xmuda: Cross-modal unsupervised domain adaptation for 3d semantic segmentation. In CVPR, 2020

---

### Decision · Program_Chairs · 2024-09-25

**Decision:**

Accept (spotlight)

**Comment:**

This paper looks at the problem of out-of-distribution detection in multi-modal settings, presenting a new benchmark MultiOOD that has a range of diverse dataset sizes and modalities stemming from five different video datasets. After benchmarking on this dataset, the authors observe that softmax predictions across the modalities are similar within distribution but disagree widely out-of-distribution, and show empirically that this tends to correlate with poor OOD performance. An Agree-to-Disagree algorithm is developed to drive agreement on the correct class and disagreement on the incorrect classes, along with outlier synthesis, showing state of art results on this benchmark.

  The reviewers all agreed that the paper introduces an interesting benchmark, has good contributions with respect to the algorithmic improvements, and is clearly written. Some clarifications and concerns were raised, including clarification of the architecture (e.g. per-frame or per-video feature usage, protocol and robustness to missing modalities, lack of comprehensiveness of baselines (e.g. ensembles), and extensibility to domains other than activity recognition. The authors provided a comprehensive rebuttal including new results that, as mentioned by the reviewers, addressed most of the concerns.

  Overall, taking into account all of the information I recommend acceptance of this paper. The benchmark is likely to be highly impactful as multi-modal models in general become ubiquitous, and the additional algorithmic contributions are both interesting, uniquely adapted to the problem, and effective. I recommend that the authors include the many additional results and exposition in the camera-ready paper.